# Null effect of perceived drum pattern complexity on the experience of groove

Olivier Senn[1]*, Florian Hoesl[1], Toni Amadeus Bechtold[1,2], Lorenz Kilchenmann[1], Rafael Jerjen[1], Maria Witek[2]

1 School of Music, Lucerne University of Applied Sciences and Arts, Lucerne, Switzerland, 2 Department of Music, University of Birmingham, Birmingham, United Kingdom

* olivier.senn@hslu.ch

**Data Availability Statement:** All data files are available from the Zenodo data repository (https://zenodo.org/records/11102731).

## Abstract

There is a broad consensus in groove research that the experience of groove, understood as a pleasurable urge to move in response to music, is to some extent related to the complexity of the rhythm. Specifically, music with medium rhythmic complexity has been found to motivate greater urge to move compared to low or high complexity music (inverted-U hypothesis). Studies that confirmed the inverted-U hypothesis usually based their measure of complexity on the rhythmic phenomenon of syncopation, where rhythms with more and/or stronger syncopation are considered to be more complex than less syncopated rhythms. However, syncopation is not the same as complexity and represents only one rhythmic device that makes music complex. This study attempts the verification of the inverted-U hypothesis independently from syncopation. It uses a new stimulus set of forty idiomatic popular music drum patterns whose perceptual complexity was measured experimentally in a previous study. The current study reports the results of a listening experiment with $n = 179$ participants, in which the inverted-U hypothesis was not confirmed. Complexity did not have any significant effect on listeners' urge to move ($p = 834$). Results are discussed in the context of the psychological model of musical groove, which offers a nuance to this null result: simple drum patterns motivate listeners to dance because they convey metric clarity; complex patterns invite dancing because they are interesting. Yet, overall, the urge to move does not seem to depend on complexity, at least in the case of idiomatic drum patterns that are typically encountered in the Western popular music repertoire.

## Introduction

The psychological study of the groove experience, understood as a *Pleasurable Urge to Move to Music* [1] (or *PLUMM* [2]) has attracted increasing scholarly interest in recent years. Most research efforts so far have been empirical and aimed at identifying factors in the music or in the listener that affect the intensity of the groove experience (for an overview, see [3]). One of the major empirical results in the field relates the intensity of the groove experience to the rhythmic complexity of the music: In 2014, Witek et al. [4] and Sioros et al. [5] found that listeners' urge to move in response to music follow an inverted U-shaped function of rhythmic

**Funding:** This research was supported by the Swiss National Science Foundation (Grant No. 100016 192398 to Olivier Senn). The funders had no role in study design, data collection and analysis, decision to publish, or preparation of the manuscript.

**Competing interests:** The authors have declared that no competing interests exist.

complexity (measured in terms of syncopation), implying that music with medium rhythmic complexity triggers a stronger urge to move compared to music with very low or very high rhythmic complexity. This result resonates with Daniel Berlyne's claim that the appreciation of art is greatest for artworks with medium complexity [6, 7]. The inverted-U result was fully replicated by several studies [8–13]. Other studies replicated the result partially, finding that high-complexity stimuli caused less urge to move than stimuli with medium complexity [14–16].

In two studies, the inverted-U relationship between stimulus complexity and the experience of groove was not confirmed: In 2018, Senn et al. [17] found a significant ($p<.001$) positive relationship between complexity and the experience of groove in 250 popular music patterns, suggesting that higher complexity is associated with greater urge to move. Yet, this relationship was so weak that it can be considered to be practically irrelevant ($R^2 = .010$). In 2022, Sioros et al. [18] found that the experience of groove not only depended on the number of syncopations that appear in a musical pattern and make it complex, but also where in a pattern they arise, thus accentuating the relevance of musical syntax for the groove experience.

In all of these studies, rhythmic complexity was operationalized on the basis of syncopation. In their 2014 study, Witek et al. [4] used 50 drum patterns from Western popular music as experimental audio stimuli. They modified Longuet-Higgins & Lee's [19] weighted measure of syncopation in monophonic rhythms from 1984 in order to develop the *index of syncopation* which quantifies how strongly the bass drum and snare drum voices of popular music drum patterns are syncopated. Witek et al.'s [4] stimuli (or selections thereof) and associated values on the index of syncopation were used in listening experiments by several later studies [9, 12, 14, 15].

The index of syncopation is a heuristic approach to measuring complexity, and its validity was only recently established: In 2023, Senn et al. [20] carried out a listening experiment, in which they empirically measured the perceived complexity of 40 popular music drum patterns (different from Witek et al.'s [4]) using a pairwise comparisons approach. The index of syncopation showed a medium to strong positive correlation with perceived complexity ($r = .701, R^2 = .491$, see [20], p. 10). Therefore, the index of syncopation can be considered to be a fairly valid measure of drum pattern complexity. Nevertheless, given that the index of syncopation has been used extensively as a measure of drum pattern complexity and that syncopation is not likely to be the only aspect that contributes to complexity, it is reasonable to carry out another replication study which investigates whether the experience of groove is an inverted-U function of perceived stimulus complexity. This study presents such a replication using the 40 drum pattern stimuli and complexity measurements of Senn et al. [20].

The inverted-U relationship between syncopation and the experience of groove motivated Vuust et al. [21, 22] to develop a theory that aims to explain this finding, namely the *predictive coding of rhythmic incongruity* theory (*PCRI*), based on the predictive coding approach to cognition [23, 24]. A core idea of predictive coding is that sentient organisms act and learn in response to surprising events. PCRI identified syncopated rhythm as a source of rhythmic surprise in music: in music perception, syncopation is a rhythmic phenomenon where listeners expect a note to be played on a metrically strong position within the bar. Instead, the note is surprisingly displaced to an earlier, weaker metric position (see also [18, 19, 25]). Researchers used neuro-scientific methods to show that listeners' central nervous system indeed responds to syncopation with neurological signs of surprise, such as the mismatch negativity reaction [21, 26–28]). In simple words, PCRI can be explained as follows: rhythms with very few syncopations create few instances of surprise. The listener is capable of predicting most events with high accuracy. Yet, there is not much that can be learned from these rhythms, so they might be considered to be boring. Conversely, rhythms with many syncopations are surprising: the listener has difficulties predicting the events, and the rhythm might be considered to be

confusing. According to Stupacher et al. [29], music of medium complexity, which is at the apex of the inverted-U curve, inhabits the "sweet spot between predictability and surprise" (p. 1) because it toes the line between too boring (little syncopation = not surprising) and too confusing (much syncopation = not predictable). Operationally, Stupacher et al. modelled the groove experience as a function of the product between the risk of prediction errors (a consequence of the surprise) and prediction certainty (a measure of predictability), see also [22].

In 2017, Witek [30] argued that synchronized body movement plays an important role in the interpretation of syncopated rhythm. By synchronizing body movement with the music, listeners create a multi-sensory representation of the musical meter (involving auditory, visual, proprioceptive and tactile modalities), and occasional syncopations will not confuse the listeners, because body movement "fills in" the missing events. Accordingly, moving in synchrony with the music allows listeners to enjoy the surprises of a syncopated rhythm without suffering the associated confusion. Consequently, movement improves our listening experience. From research on multimodal perception (e.g. [31]), we can assume that synchronized body movement (resp. dancing) allows listeners to process rhythmic complexity more easily and enjoy the combined listening/dancing experience more than the listening experience alone.

The current study not only aims at replicating the inverted-U relationship between complexity and the groove experience. It also investigates whether Stupacher et al.'s [29] "sweet spot" idea can be confirmed from a behavioural perspective. This investigation will be carried out within the analysis framework of the *psychological model of musical groove* [32, 33]. The groove model is a theory and analysis framework that aims at explaining under which circumstances music listeners experience an urge to move in response to music (**Fig 1**). The model takes into account properties of the music, the personal background of the listeners, their concrete listening situation, body movement itself, and a variety of emotions and cognitive processes that are potentially connected to the urge to move. The model hypothesizes that listeners' *inner representation of temporal regularity* (REG), their *time-related interest* in the music (INT), their listening *pleasure* (PLE) and *energetic arousal* (ENE) in response to the music are all positively associated with listeners' *urge to move* (MOV). In short, the model claims that music with a strong beat (REG) and an interesting rhythm (INT) that gives us energy (ENE) and pleasure (PLE) is also likely to give us the motivation to move (MOV).

Recently, a structural equation model (SEM, [34–36]) implementation of the psychological groove model has been developed [33]. The SEM allows to test model hypotheses that imply mediation. Specifically, the model hypothesizes that musical, personal, and situational factors indirectly affect the urge to move (MOV, see arrows in **Fig 1** and in [33]) mediated through the other four cognitive processes (REG, INT, PLE, ENE). An essential part of the SEM implementation is a series of psychometric scales that were developed in recent years and allow to measure the intensity of music listeners' experience of PLE and MOV [37] as well as REG, INT, and ENE [38].

Two of the mediation pathways are of particular importance for the current study, because they allow to express the "sweet spot" idea in terms of the psychological groove model. These are the pathways that measure the indirect effects of drum pattern complexity (a musical property, left box in **Fig 1**) on MOV mediated through the REG and INT latent variables:

- The experience of temporal regularity (REG) can be understood as a subjective response to the predictability of the rhythm. More complex patterns are likely to be heard as less regular, so we can expect complexity and REG to be negatively associated. Since REG is thought to be positively associated with MOV, we can expect drum patterns with high complexity to provoke an impression of low regularity (REG) in the listener and thus, indirectly affect MOV negatively (see $H_2$, below).

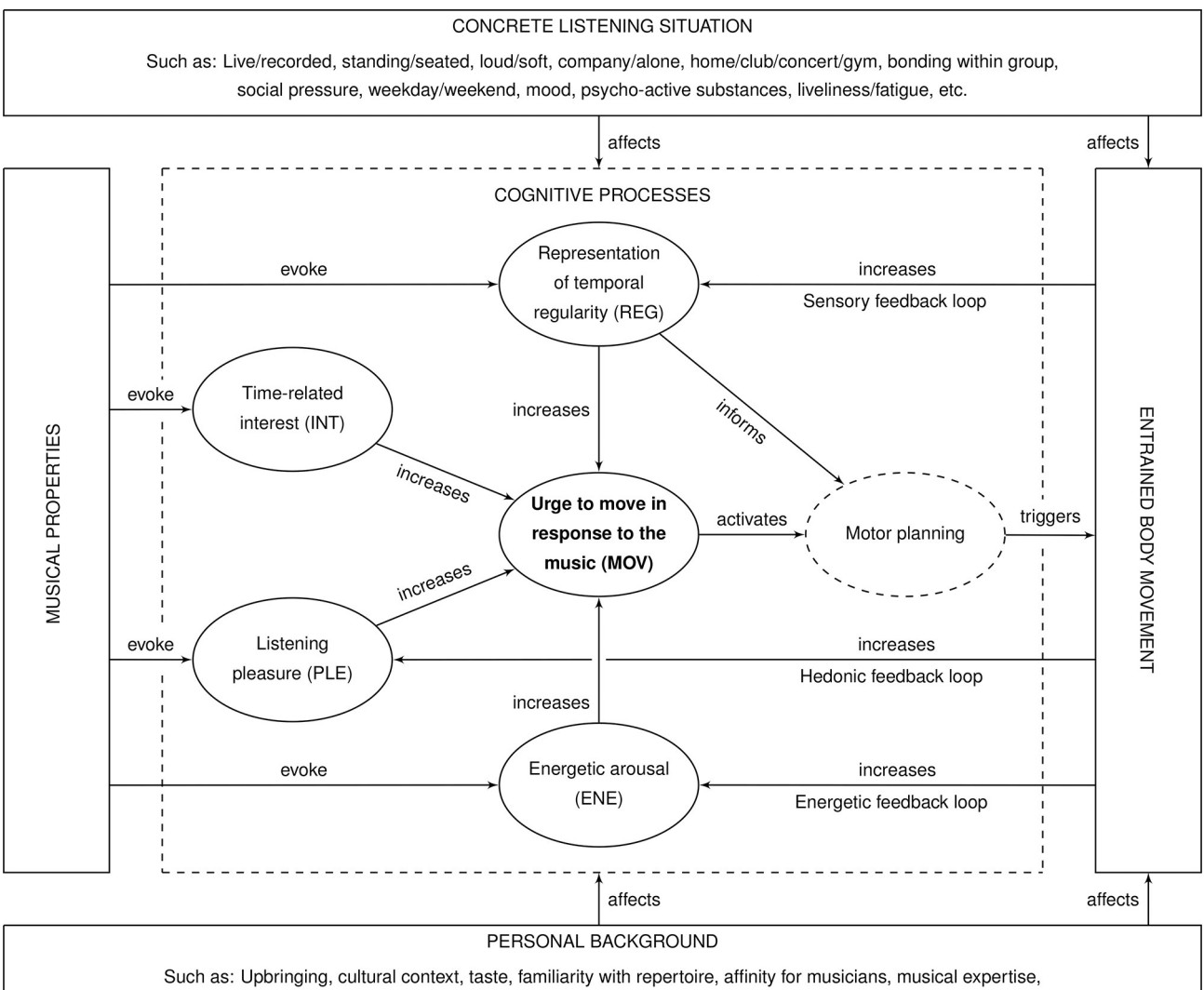

**Fig 1. Psychological model of musical groove.** Version 2.3, adapted from [32, 33].

- Further, we can assume that the rhythmic surprises of complex drum patterns catch the attention of the listeners and attract their interest (*INT*) to the rhythm, so we expect that complexity and *INT* are positively correlated. According to the groove model, *INT* has a positive effect on *MOV*. As a consequence, the indirect effect of drum pattern complexity on *MOV*, mediated through *INT*, can be expected to be positive ($H_3$, below). Conversely, very simple drum patterns will be heard as boring (low *INT*) and for this reason be associated with low *MOV*.

  The pathways from complexity to *MOV* through *REG* and *INT* implement the "sweet spot" idea within the groove model: drum patterns of medium complexity motivate strong *MOV*, because they are complex enough not to be boring, and regular enough not to be confusing. If the inverted-U relationship between complexity and the urge to move can be confirmed, the two indirect effects through *REG* and *INT* may offer a psychological explanation why the inverted-U arises.

In summary, this study aims to replicate the main result of Witek et al. [4] that listeners' urge to move in response to popular music drum patterns is an inverted-U shape of the patterns' complexities. This investigation is based on a new set of forty drum pattern stimuli combined with an empirical measure of perceived complexity that is not based on syncopation [20]. The study results can be considered to be independent from those of the 2014 Witek et al. study [4], and from all replications that confirmed the inverted-U hypothesis, because none of the materials and measures from these previous studies were re-used. We formulate the following three hypotheses:

- $H_1$: Listeners' urge to move (*MOV*) is strongest for drum patterns with medium complexity, compared to low or high complexity (inverted-U hypothesis).

- $H_2$: Stimulus complexity negatively affects listeners' impression of temporal regularity (*REG*), which indirectly leads to a reduced urge to move (*MOV*).

- $H_3$: Stimulus complexity positively affects listeners' interest in the rhythm of the drum pattern (*INT*), which indirectly increases their urge to move (*MOV*).

The three hypotheses are complementary: $H_1$ allows to test the inverted-U hypothesis on a new stimuli set, replacing syncopation by perceived complexity. $H_2$ and $H_3$ identify potential psychological mechanisms that implement the inverted-U relationship (and thus the "sweet spot" idea) within the groove model.

## Methods and materials

### Stimuli

The audio stimuli for this experiment were created for an earlier study [20] and are publicly available (https://zenodo.org/records/10728042). They consist of 40 drum patterns from Western popular music, meticulously transcribed and reconstructed from commercial recordings that were originally played by renowned and famous drummers (e.g. John Bonham, Roger Taylor, Bernard Purdie, and others) in the context of full-band recordings. The patterns are a selection from the *Lucerne Groove Research Library* (250 drum patterns, https://www.grooveresearch.ch/, see also [17]). The selection of the forty drum patterns aimed at plausibly covering the complexity range that is idiomatic for the Western popular music drum pattern repertoire. The appendix (Table 5 in S1 Appendix) offers some discographic information about the original full-band recordings from which the drum patterns were extracted. More details about drum pattern selection are given in [20].

The instrumentation of each drum pattern consists of the bass drum, the snare drum and one or more cymbals (hi-hat, ride cymbal, crash cymbal, etc.). Each pattern is presented during four bars (plus the first beat of bar 5). The audio was rendered from MIDI using high-quality drum samples from a sound library (Toontrack Superior Drummer Custom & Vintage, version 2.4.4). The reconstructions not only represent the syntax of the patterns (notes, rests), but also microtemporal variations in the millisecond range (based on computer-assisted measurements), and different dynamics/loudness levels (based on transcribers' judgement).

Each stimulus is associated with an empirical value (**Table 1**) of perceived complexity that was measured previously in a listening experiment with 220 participants from predominantly Western countries [20]. The experiment had a pairwise comparison design: in each of the $n = 4400$ trials, participants listened to two drum pattern stimuli and decided which of the two sounded more complex to them. Each participant carried out 20 trials which presented all 40 patterns exactly once. The Bradley-Terry probability model was used to analyze the comparison data and to estimate a perceived complexity coefficient (real numbers in the range [0.400, 4.701]).

**Table 1. Drum pattern stimuli with perceived complexity, *MOV*, *REG*, *INT*, *PLE*, *ENE*, and *FAM* ratings.**

| No. | Song Title (Group) | Perceived Complexity | *MOV* | *REG* | *INT* | *PLE* | *ENE* | *FAM* |
|-----|--------------------|----------------------|-------|-------|-------|-------|-------|-------|
| 1 | A Kind Of Magic (A) | 0.400 | 4.136 | 5.227 | 3.742 | 4.083 | 4.330 | 2.432 |
| 2 | (Sittin' On) The Dock Of The Bay (B) | 0.408 | 3.030 | 5.094 | 3.037 | 3.356 | 2.761 | 2.333 |
| 3 | Smells Like Teen Spirit (C) | 0.476 | 3.636 | 5.128 | 3.124 | 3.364 | 3.238 | 2.814 |
| 4 | Boogie Wonderland (D) | 0.573 | 3.681 | 4.628 | 3.362 | 3.766 | 4.277 | 2.553 |
| 5 | Vultures (A) | 0.784 | 3.212 | 5.335 | 2.765 | 3.379 | 3.097 | 2.341 |
| 6 | Kashmir (B) | 1.182 | 2.519 | 4.789 | 2.548 | 2.874 | 1.906 | 2.333 |
| 7 | Street Of Dreams (C) | 1.210 | 3.047 | 4.558 | 2.605 | 3.240 | 2.221 | 2.395 |
| 8 | Change The World (D) | 1.230 | 2.950 | 4.628 | 2.362 | 2.993 | 2.324 | 2.234 |
| 9 | Let's Dance (A) | 1.263 | 3.598 | 4.432 | 3.795 | 4.091 | 3.801 | 2.000 |
| 10 | Space Cowboy (B) | 1.415 | 3.785 | 4.672 | 3.452 | 3.889 | 3.328 | 2.356 |
| 11 | I Feel For You (C) | 1.632 | 4.039 | 4.203 | 3.349 | 3.922 | 3.610 | 2.860 |
| 12 | Virtual Insanity (D) | 1.859 | 3.149 | 4.878 | 2.851 | 3.362 | 2.702 | 2.277 |
| 13 | Bravado (A) | 1.902 | 3.955 | 5.057 | 4.068 | 4.303 | 4.636 | 2.227 |
| 14 | Let's Go Dancing (B) | 1.951 | 3.015 | 4.456 | 2.889 | 3.341 | 2.800 | 2.133 |
| 15 | Discipline (C) | 2.091 | 4.411 | 4.924 | 3.682 | 4.023 | 4.262 | 2.674 |
| 16 | Pass The Peas (D) | 2.151 | 3.099 | 4.840 | 2.759 | 3.234 | 2.676 | 2.234 |
| 17 | The Pump (A) | 2.189 | 3.167 | 4.142 | 3.606 | 3.864 | 3.432 | 1.932 |
| 18 | Roxanne (B) | 2.216 | 4.415 | 4.217 | 4.533 | 4.430 | 4.761 | 2.711 |
| 19 | Dreamin' (C) | 2.413 | 3.736 | 4.779 | 3.233 | 3.566 | 3.570 | 2.395 |
| 20 | Soon I'll Be Loving You Again (D) | 2.511 | 3.213 | 4.617 | 3.574 | 3.468 | 2.670 | 2.170 |
| 21 | Summer Madness (A) | 2.530 | 2.924 | 4.642 | 3.356 | 3.508 | 2.386 | 1.886 |
| 22 | Listen Up! (B) | 2.586 | 3.156 | 4.694 | 3.185 | 3.244 | 2.761 | 2.200 |
| 23 | Jungle Man (C) | 2.752 | 3.023 | 4.384 | 3.519 | 3.519 | 2.651 | 2.163 |
| 24 | Shake Everything You Got (D) | 3.051 | 2.631 | 4.016 | 2.950 | 3.085 | 2.761 | 2.362 |
| 25 | Chicken (A) | 3.052 | 4.098 | 4.023 | 4.758 | 4.629 | 4.284 | 2.295 |
| 26 | Cissy Strut (B) | 3.080 | 2.496 | 4.067 | 2.874 | 2.830 | 2.017 | 2.067 |
| 27 | Far Cry (C) | 3.120 | 4.349 | 3.977 | 4.209 | 4.147 | 4.680 | 2.512 |
| 28 | Alone + Easy Target (D) | 3.130 | *n.d.* | 3.250 | 3.957 | 3.901 | 4.564 | 2.191 |
| 29 | Soul Man (A) | 3.263 | 4.015 | 4.534 | 4.477 | 4.485 | 4.523 | 2.159 |
| 30 | Ain't Nobody (B) | 3.300 | 3.837 | 4.583 | 4.422 | 3.933 | 4.556 | 2.311 |
| 31 | Diggin' On James Brown (C) | 3.314 | 2.899 | 3.291 | 3.868 | 3.287 | 3.058 | 1.721 |
| 32 | In The Stone (D) | 3.342 | 3.553 | 3.750 | 4.007 | 3.603 | 3.537 | 2.043 |
| 33 | Southwick (A) | 3.360 | 3.273 | 3.790 | 4.136 | 4.205 | 3.591 | 2.114 |
| 34 | You Can Make It If You Try (B) | 3.447 | 2.970 | 3.489 | 3.430 | 3.489 | 3.667 | 1.978 |
| 35 | The Dump (C) | 3.464 | 3.690 | 4.000 | 4.047 | 3.876 | 3.209 | 2.023 |
| 36 | Killing In The Name Of (D) | 3.564 | 3.858 | 4.165 | 4.078 | 3.546 | 4.633 | 2.298 |
| 37 | Cold Sweat (A) | 3.763 | 2.932 | 3.085 | 3.894 | 3.553 | 3.136 | 1.909 |
| 38 | Hyperpower (B) | 3.902 | 3.244 | 3.583 | 4.459 | 3.837 | 4.244 | 2.156 |
| 39 | Rock Steady (C) | 4.394 | 3.589 | 3.157 | 4.326 | 3.543 | 4.384 | 2.233 |
| 40 | Jelly Belly (D) | 4.701 | 3.539 | 2.457 | 4.014 | 3.390 | 3.633 | 2.021 |

*Notes*: Perceived Complexity: (see [20]), *MOV*: urge to move, *REG*: representation of temporal regularity, *INT*: time-related interest, *PLE*: listening pleasure, *ENE*: energetic arousal, *FAM*: familiarity. For stimuli metadata see Appendix (Table 5 in S1 Appendix). Stimuli are listed by ascending perceived complexity value. For one stimulus (28, Alone + Easy Target) *MOV* ratings were not correctly recorded during the experiment (*n.d.*). This stimulus was subsequently excluded from the analysis. Group: Stimuli were divided into four experimental groups (A-D) that were judged by different participant sub-samples.

The perceived complexity coefficients have an intuitive probabilistic interpretation: for each pair of stimuli, they allow to calculate the probability that one stimulus will be considered to be more complex than the other in a pairwise comparison trial when judged by a member of the surveyed listener population. To make an example, the most complex stimulus in the set (No. 40, "Jelly Belly", perceived complexity = 4.701) has the following estimated probability $\hat{\Pi}_{40,1}$ of winning a pairwise comparison trial against the least complex stimulus (No. 1, "A Kind of Magic", perceived complexity = 0.400):

$$\hat{\Pi}_{40,1} = \frac{e^{4.701-0.400}}{1 + e^{4.701-0.400}} \simeq .987.$$

"Jelly Belly" has a very high probability to win a pairwise comparison trial against "A Kind of Magic" when judged by a member of the listener population. The formula for $\hat{\Pi}$ uses the logistic function which, in this case, takes parameter 4.701–0.400. For more information about the perceived complexity measure and the associated calculations, please refer to [20] (pp. 5–7). Note that this measure differs substantially from the index of syncopation: it is based on listeners' subjective statements in response to the stimulus, whereas the index of syncopation is based exclusively on the objective rhythmic properties of the patterns.

## Measures

During the experiment, we collected data on the following variables to assess aspects of participants' personal background:

- *Demographic information*: Year of birth, gender, country of residence, English language skills, musical expertise (professional musician, music student, amateur musician, music listener, not interested in music), participants' everyday use of music, and the characteristics of music they like.

- *Popular music affinity*: We used an adaptation of the Short Test of Music Preferences [17, 39] questionnaire to collect preference data on 21 musical styles. The *popular music affinity* scale was calculated as the mean of participants' ratings on 14 popular music styles (pop, disco, rock, blues, country/western, rock'n'roll, heavy metal, rhythm & blues, soul, funk, rap/hip hop, dance/electronic, alternative/indie, reggae/dub/dancehall), and had a range of [0,6], see also [33].

- *Musical training*: Musical training subscale of the English-language Goldsmiths Music Sophistication Index [40] with seven items and a range of [7,49].

- *General affinity to dance*: The urge to dance scale of the Goldsmiths Dance Sophistication Index [41] with 5 items and a range of [5,35]. This scale measures a personality trait, namely how much respondents like to dance to music in general.

Participants provided additional information about their daily use of music and the characteristics of the music they like. This data will be analyzed in a later study on music preferences. Participants' assessed the stimuli with respect to five dimensions that will be presented subsequently. All dimensions were measured with composite scales based on Likert-type items with seven answer categories that indicate levels of agreement (strongly disagree, disagree, slightly disagree, neither agree nor disagree, slightly agree, agree, strongly agree):

- *Urge to move* (*MOV*): Participants' urge to move in response to a musical stimulus with three items and range [0,6], first presented in [37]. The *MOV* scale measures participants' short-term response to a specific stimulus. It should not be confunded with the DSI's urge to

dance scale, which measures a personality trait and respondents' general inclination to dancing.

- *Listening pleasure* (*PLE*): Participants' enjoyment while listening to a musical stimulus with three items and range [0,6], see [37].

- *Representation of temporal regularity* (*REG*): This scale measures to what extent listeners feel that a musical stimulus gives them an experience of temporal regularity. It has four items and a range of [0,6], see [38].

- *Time-related interest* (*INT*): This scale with three items and range [0,6] assesses to what extent the rhythm of a musical stimulus is interesting to the listeners, see [38].

- *Energetic arousal* (*ENE*): This scale (four items, range [0,6]) measures how strongly listeners feel energized by a musical stimulus, see [38].

Additionally, participants indicated whether they were familiar with the music:

- *Familiarity* (*FAM*): Participants answered to the question "Have you heard this music in the past?" in response to a musical stimulus (definitely not, probably not, I do not know, probably yes, definitely yes). This single-item scale had a range of [0,4] and was already used in [33].

Items for all measurement scales are presented in the Appendix (Table 6 in S1 Appendix). The numerical *perceived complexity* of the 40 drum pattern stimuli was the only stimulus-related variable used in the analysis. It was measured in [20] with a different sample of participants.

## Participants

A preliminary power analysis was carried out to investigate how many participants were required to detect a small effect of $R^2_{adj.} = .02$ with respect to the first hypothesis. $H_1$ will be confirmed if the quadratic term of a *MOV ~ Complexity+Complexity²* model is significantly smaller than zero at the $\alpha = .05$ significance level, and if the apex of the quadratic function is located within the complexity range of the stimuli (thus indicating an inverted-U function). Based on the data of an earlier study [33], we assumed that the urge to move (*MOV*) ratings will have an error variance of approximately $\sigma^2 = 1.9$, and that the individual differences between the drum patterns will have a large effect on *MOV* ratings ($R^2_{adj.} = .14$), independent of complexity. The power analysis was programmed in *R* using a Monte Carlo method with 5000 simulated samples. The simulation reached a power of $\beta = .80$ with $n = 150$ participants, where each participant rated 10 stimuli ($N = 1500$ observations in total).

Participants were recruited through the platform *Prolific* (https://www.prolific.com/) The data of 180 participants was collected, yet the data of one participant was incomplete. The remaining 179 participants passed a systematic investigation of streamlining and rushing. Participants reported high English language skills: 74 were native speakers, 86 indicated competent language use (C1 or C2), 16 indicated independent (B1 or B2), and 3 basic language use (A1 or A2). In the case of the 19 participants with A or B language levels, their answers to a free text response were checked for intelligibility and coherence. All 19 gave sensible answers and were thus included in the analysis.

Participants (76 female, 98 male, 3 other, 2 no answer) had a mean age of 32.8 years ($SD = 11.6$). They lived in a variety of countries: United Kingdom (37), South Africa (27), Poland (25), Portugal (17), Italy (11), Spain (8), Canada (6), Hungary (6), Mexico (6), Netherlands (6), Australia (4), Czech Republic (4), and others (20). Two participants did not provide information on their country of residence.

Most participants self-identified as music listeners (155) and some as amateur musicians (19). Three participants identified as professional musicians, one as a music student, and one person indicated not to be interested in music. The sample of participants had a very low mean score of 15.49 (*SD* = 8.81) on the Gold MSI's Musical Training scale ($t_{(178)} = -16.746, p < .001$, Cohen's *d* = −1.25), compared to a normative UK population with a mean score of 26.52 ([40], p. 10).

## Procedure

Participants filled the questionnaire online on the SosciSurvey platform (www.soscisurvey.de) on November 16, 2023. They were asked to seek a quiet room and to use quality headphones in order to carry out the experiment. They were informed about the general purpose of the study, the use of the data, and gave informed consent by mouse click. They provided demographic information about themselves and gave information about their music preference and use. Participants listened to a test drum pattern audio stimulus (similar to the experimental stimuli but not identical with any of them) in order to adjust playback volume to a loud but agreeable intensity. They were asked not to change the volume during the experiment.

Participants were randomly assigned to one of four participant groups (Group A: *n* = 44 participants, B: *n* = 45; C: *n* = 43; D: *n* = 47). Each group judged a subset of ten stimuli from the entire set of 40. Subsets were selected such that the ten stimuli cover a wide range of complexity (see **Table 1**, "Group"). Experimental data was collected in five blocks. In the first block, participants judged the urge to move triggered by the stimuli (*MOV*) and their familiarity with the music (*FAM*). In each of blocks 2–5, participants listened to the stimuli again and rated to what extent the music evoked an inner representation of temporal regularity (*REG*); time-related interest (*INT*); listening pleasure (*PLE*); and energetic arousal (*ENE*). The *MOV* & *FAM* block was always presented first: the *MOV* scale was the main response variable of the study. Accordingly, we wanted participants to fill it early in the experiment at a moment of maximum motivation. This also made sure that participants responded to the *FAM* item upon hearing the stimulus for the first time in the experiment. The sequence of blocks 2–5 (*REG*, *INT*, *PLE*, *ENE*) was randomized. The sequence of the ten stimuli in each block and the presentation order of the 3 or 4 items of the composite scales were also randomized. Between experimental blocks, participants filled the items relating to the DSI's urge to dance scale [41] and the musical training scale of the Goldsmiths Music Sophistication Index [40]. At the end of the experiment, participants could read a short text on the goals of the study and on the hypotheses it aims to test. Participants used a median of 27 minutes to fill the survey.

The collection of the experimental data in five different blocks was time-consuming for participants, because they needed to listen to and rate every stimulus five times. In order to keep the experiment at a duration of approximately 30 minutes, we had to reduce the number of drum patterns to ten per participant. The objective of the five-blocks setup was to decouple the five latent variables relevant to the groove experience, which in previous studies [33, 37, 38, 42] were highly positively correlated, as they were collected in the same trial.

## Analysis

Due to a programming error in the online survey, the *MOV* data of one stimulus in group D (No. 28, Alone + Easy Target) was not correctly collected. All data (*n* = 47 observations) relating to this stimulus were discarded prior to the analysis. The final dataset for analysis consisted of ($N = 179 \times 10 - 47 = 1743$) complete observations on 39 stimuli with ratings on *FAM* and on the *MOV*, *REG*, *INT*, *PLE*, and *ENE* latent variables. Additionally, we used listeners'

familiarity with the music (*FAM*), their score on the Gold-MSI training scale (*MSI*), their Gold-DSI urge to dance score (*DSI*), their age and gender in the analysis.

In order to test $H_1$, a quadratic regression model was fitted to the data, predicting the urge to move (*MOV*) ratings from the complexity and the squared complexity of the stimuli. With respect to $H_2$ and $H_3$, we fit a structural equation model to the data that adapts the groove model framework [32, 33] to this specific study design and allows to carry out the mediation analysis necessary to test the study hypotheses. The SEM analysis involved perceived complexity (as the only musical property), all latent variables (*MOV*, *REG*, *INT*, *PLE*, and *ENE*), familiarity (*FAM*), Gold-MSI training scale (*MSI*), Gold-DSI urge to dance scale (*DSI*), popular music affinity, gender, and age. Statistical analyses were carried out in *R* (4.3.0) within the *RStudio* (2023.09.1+494) environment. For structural equation modelling, *lavaan* (0.6–15) was used. The experimental data is available under https://zenodo.org/records/11102731.

## Ethics statement

The ethics commission of the Lucerne University of Applied Sciences Arts approved the design of this study on December 15, 2022 (decision letter EK-HSLU 004 M 22). The study was carried out according to the principles of the Declaration of Helsinki; it was not preregistered.

## Results

### $H_1$: Inverted-U hypothesis

Participants' urge to move (*MOV*) ratings were regressed on the complexity and squared complexity measures associated with the drum patterns. The quadratic regression model did not explain a significant proportion of the variance in the *MOV* ratings ($p$ = .834). We found no inverted-U relationship between complexity and *MOV* since the squared complexity coefficient in the model was not significantly lower than zero (see **Table 2** and **Fig 2**). Consequently, $H_1$ was not confirmed: the experiment provided no evidence that listeners' urge to move is strongest for drum patterns with medium perceived complexity, compared to low or high perceived complexity.

We removed the quadratic term from the model in order to investigate whether there is a significant linear relationship between complexity and the *MOV* ratings, but none was found ($p$ = .857). As a manipulation check, we tested whether the experimental stimuli had any effect on the *MOV* ratings all. We found a medium-sized effect of the stimuli on the *urge to move* ($MOV, F_{(38,1704)} = 5.648, p < .001, R^2_{adj.} = .092$, with estimated error variance $\hat{\sigma}^2 = 2.12$).

Earlier results on the relationship between complexity and the groove experience were based on syncopation as a measure of drum pattern complexity. Syncopation estimates were calculated for the experimental stimuli (index of syncopation [4] in the formalisation of [43]), in order to test whether the *MOV* ratings are an inverted-U function of syncopation instead of perceived complexity. The drum patterns showed a range of [0.5, 8.3] units per beat on the index of syncopation. The 50 Witek et al. [4] stimuli had a wider range on the index of

**Table 2. Quadratic model predicting *urge to move* (*MOV*) from *complexity* and *complexity*$^2$.**

| Source | Estimate | SE | t | p |
|---|---|---|---|---|
| Intercept | 3.507 | 0.142 | 24.670 | < .001 |
| Complexity | −0.078 | 0.130 | −0.603 | .547 |
| Complexity$^2$ | 0.016 | 0.027 | 0.576 | .565 |

*Notes*: SE: standard error of the estimate; t: t-statistic; p: significance probability.

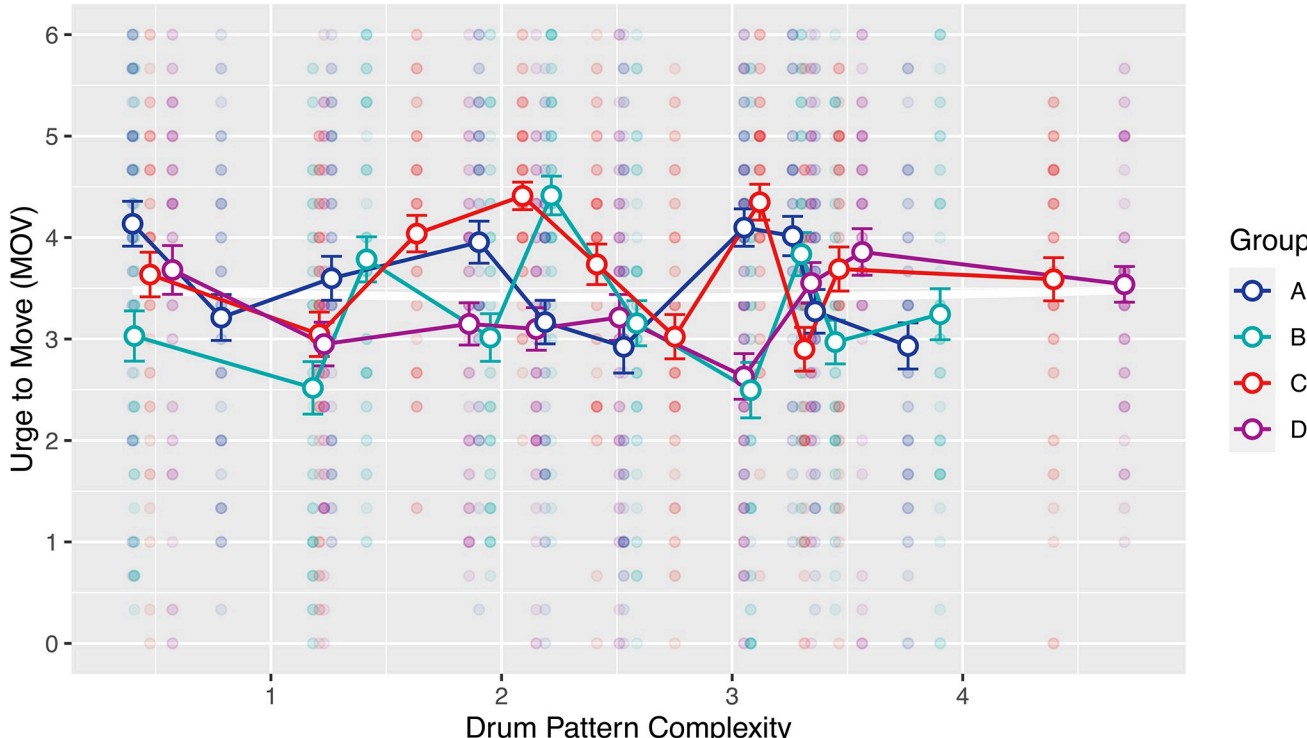

**Fig 2. Scatterplot of urge to move (*MOV*) ratings in function of perceived drum pattern complexity.** *The foreground represents mean ratings per stimulus (error bars are the standard error of the mean) in the four experimental groups (A-D). Small semitransparent symbols in the background are single MOV ratings. The white curve in the background is the quadratic function specified by the model of* **Table 2**.

syncopation of [0.0, 10.1] units per beat, exceeding this study's range by nearly 30%. The syncopation per beat measure takes into account the different lengths of drum patterns between two bars (Witek et al. [4]) and four bars (this study). Note, that the index is only based on syncopation in the bass drum and snare drum voices. The cymbals are not considered.

We fit a quadratic model regressing this study's *MOV* ratings on syncopation (per beat) and squared syncopation (per beat). The model showed a significant negative linear ($p = .001$) and a significant positive quadratic coefficient ($p = 0.011$) with the apex of the parabola at 4.84 syncopation per beat. The quadratic model explained only a very small proportion of the variance ($R^2_{adj.} = .006$). The model did not support the inverted-U hypothesis, since the quadratic term was positive and the apex of the function was therefore a minimum. Consequently the U was not inverted, as can be seen in **Fig 3**.

### $H_2$: Mediation through the temporal regularity (*REG*) node

A structural equation model [SEM, 34–36] was fitted to the data using the implementation methodology of [33]. Drum pattern complexity was the only stimulus-related exogenous variable in the model. Further exogenous variables were related to the person of the participant: familiarity with the stimulus (*FAM*), MSI music training, DSI urge to dance, popular music affinity, gender, and age (see **Fig 4**). The model had a very good fit with a RMSEA at .037 (90% CI: .034, .040) and a CFI at .983 (for further fit statistics, see **Fig 4**). The latent variables showed high reliabilities with Cronbach's $\alpha = .86$ (*REG*), .88 (*MOV*), .90 (*INT*), .96 (*PLE*), and .96 (*ENE*). These reliability statistics were similar to the ones found in [33]. The model in **Fig 4** reports standardized regression coefficients: they indicate the expected change in the response

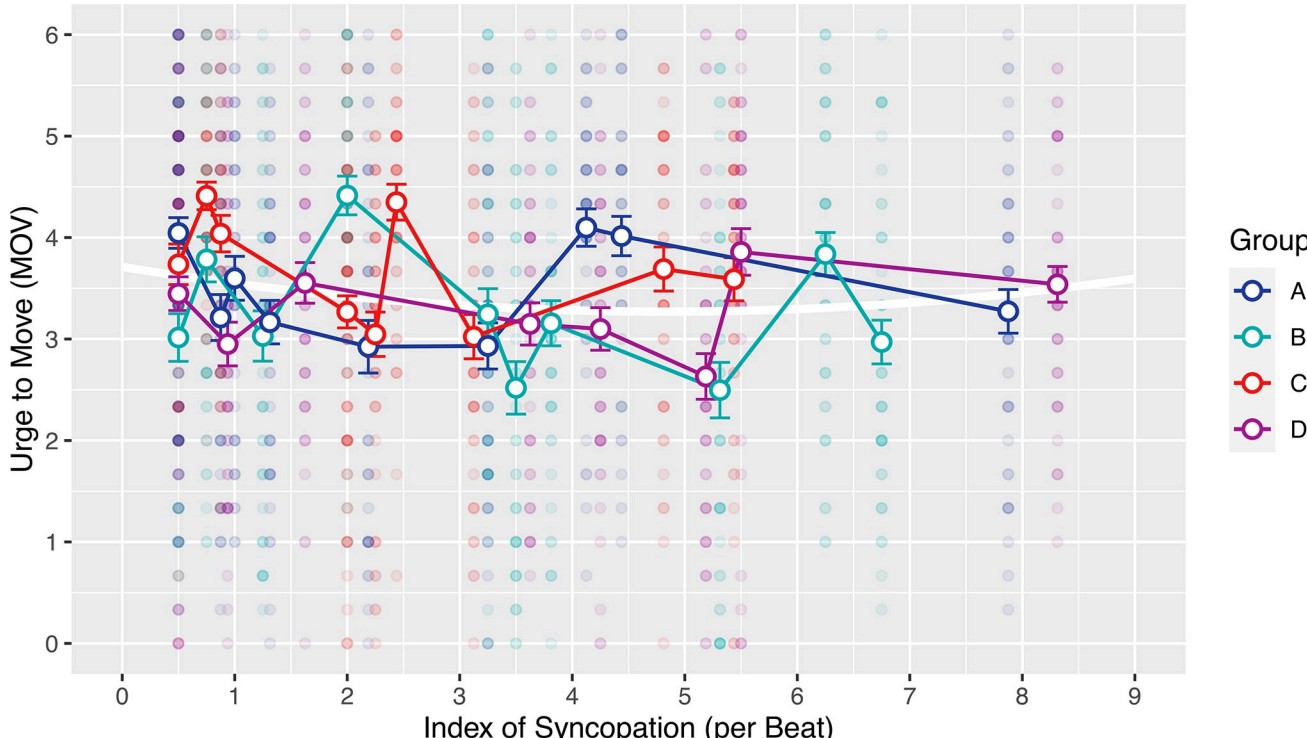

**Fig 3. Scatterplot of urge to move (*MOV*) ratings in function of syncopation (per beat).** The foreground represents mean *MOV* ratings per stimulus (error bars are the standard error of the mean) in the four experimental groups (A-D). Small semitransparent symbols in the background are single *MOV* ratings. The white curve in the background is the parabola specified by the quadratic model.

variable (measured in standard deviations) associated with a change of one standard deviation in the predictor variable.

We found a strong and highly significant negative effect of drum pattern complexity on listeners' experience of temporal regularity (*REG*). Listeners reported that the more complex drum patterns sounded less regular to them than the simpler ones ($b_{Comp \rightarrow REG} = -.408$, **Fig 4**). In turn, stimuli with higher perceived *REG* were associated with higher *urge to move* ($b_{REG \rightarrow MOV} = .066$). We combine the two regressions and observe a significant ($p = .005$) indirect negative effect of drum pattern complexity on *MOV* mediated through *REG*, which amounts to $b_{comp \rightarrow REG \rightarrow MOV} = -0.027$. This confirms $H_2$: Stimulus complexity affects listeners' impression of temporal regularity negatively, which in turn leads to reduced urge to move.

### $H_3$: Mediation through the time-related interest (*INT*) node

We found a strong and highly significant positive effect of drum pattern complexity on listeners' experience of time-related interest (*INT*): listeners reported that the more complex drum patterns were rhythmically more interesting to them than the simpler ones ($b_{Comp \rightarrow INT} = 0.263$, see **Fig 4**). Rhythmic interest in turn had a positive effect on the *urge to move* ($b_{INT \rightarrow MOV} = .128$). The indirect effect of complexity mediated through *INT* was highly significant ($p < .001$) and had a size of $b_{Comp \rightarrow INT \rightarrow MOV} = 0.034$. This confirms $H_3$: drum pattern complexity positively affects listeners' rhythmic interest (*INT*), which in turn increases the urge to move (*MOV*).

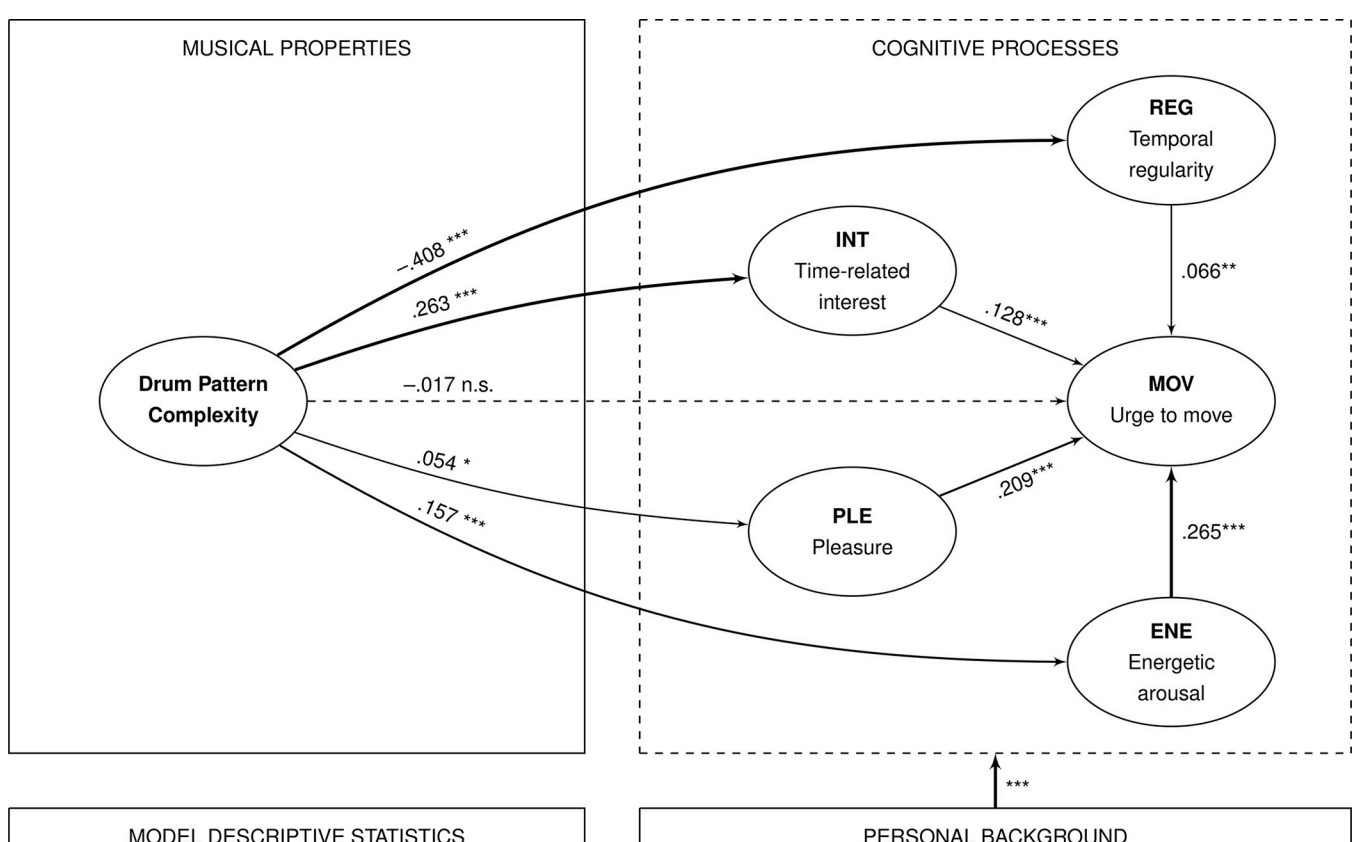

**Fig 4. Structural equation model (SEM) to test hypotheses $H_2$ and $H_3$.** All regression coefficients were standardized. The measurement model of the five latent variables is presented in the Appendix (Table 7 in S1 Appendix). Significance levels: n.s. (not significant): $p \geq .050$; *: $.010 \leq p < .050$; **: $.001 \leq p < .010$; ***: $p < .001$.

### Other effects and mediation pathways

Unrelated to this study's hypotheses, the model shows further effects: there was a small positive indirect effect of complexity on the urge to move, mediated through listening pleasure $(b_{Comp \rightarrow PLE \rightarrow MOV} = 0.011, p = .021)$. This means that more complex patterns are considered to be more pleasurable to listen to, which in turn has a positive effect on the urge to move. Also, there is a positive effect mediated through energetic arousal $(b_{Comp \rightarrow ENE \rightarrow MOV} = 0.042, p < .001)$: more complex drum patterns are better at energizing the listeners compared to simple patterns, which in turn increases the urge to move.

Of the listener-related personal background variables, familiarity (*FAM*) had a positive effect on all five latent variables and thus directly and indirectly on the urge to move.

Cumulatively, the effect from *FAM* on *MOV* amounted to $b_{FAM\to\cdots\to MOV} = 0.403$ (**Table 3**). Further, listeners' affinity with Western popular music (pop music affinity) had a positive effect on the interest (*INT*) and pleasure (*PLE*) experienced by the listeners and thus indirectly affected the urge to move (*MOV*) with a positive net effect of $b_{POP\to\cdots\to MOV} = 0.159$.

There was a positive effect of the Gold-DSI urge to dance scale on *INT*, *PLE*, *ENE*, and *MOV*, indicating that people who in general like dancing also found the drum pattern stimuli more interesting, pleasurable, energizing and movement-inducing, compared to those who do not like to dance. Cumulatively, the direct and indirect DSI effects on *MOV* amounted to $b_{DSI\to\cdots\to MOV} = 0.126$. The Gold-MSI musical training scale was not related to any of the latent variables, so musical training does not seem to play a role within the model.

**Table 4** shows how indirect effects are transmitted to *MOV* through the four mediators. The most important mediator in the model was *ENE*, which channels a total standardized indirect effect of 0.161 from all exogenous variables, followed by *PLE* (0.109) and *INT* (0.093). Effects mediated through *REG* nearly cancelled each other out (–0.010).

## Discussion

This study investigated the relationship between the perceived complexity of drum patterns from Western popular music (as measured in [20]) and the feeling of wanting to move in response to the music (*MOV*). A majority of previous studies found that listeners' urge to move forms an inverted U-function of stimulus complexity, where complexity was operationalized on the basis of syncopation [4, 5, 8–13]. Others replicated the result only partly [14–16] or not at all [17, 18]. The present study failed to confirm the inverted-U hypothesis: a quadratic regression model did not provide evidence for $H_1$ that medium complexity drum patterns are associated with greater *MOV* ratings, compared to simple or highly complex patterns.

The other two hypotheses were confirmed: perceived complexity does have a negative effect on the experience of temporal regularity (*REG*), which indirectly affects the urge to move (*MOV*) in a negative way ($H_2$). Conversely, complex patterns were perceived as more interesting, which indirectly had a positive effect on the urge to move ($H_3$). These results suggest that both simple and complex drum patterns can invite listeners to dance, but they do so for different reasons: simple patterns are suitable for dancing, because the beat and the meter are easy to detect ($H_2$); and more complex patterns are suitable for dancing, because they are interesting ($H_3$), but also because they are pleasurable and give us energy (**Fig 4**).

### A comparison of stimuli sets across studies

Why was the inverted-U hypothesis ($H_1$) not confirmed in this study? To better understand this result, we take a closer look at the stimuli used for the current investigation, compared to the stimuli used in studies that confirmed the inverted-U hypothesis.

This study's audio stimuli are drumset patterns that have been transcribed from Western popular music full band recordings in pop, rock funk styles, and reconstructed on a MIDI basis using high-quality audio samples. Senn et al. [20] made their best effort to create audio stimuli that are ecologically valid replicas of the drum patterns originally played in the full band recording. All of these drum pattern stimuli can be considered to be idiomatic within their repertoire. The 40 stimuli have been selected from a corpus of 250 drum patterns with the explicit intention to cover the complexity range typically encountered in drum patterns of the pop, rock, and funk repertoires (see arguments provided in [20]).

How do this study's stimuli compare to the stimuli used in earlier studies that successfully replicated the inverted-U hypothesis? Of the 50 drumset stimuli used in 2014 by Witek et al. [4], 36 were modelled on drum patterns from original musical contexts (such as popular music

**Table 3. Direct and mediated effects of the exogenous variables on the urge to move (*MOV*).**

| Exogenous variable | Regression path | Standardized Estimate (*b*) | *p* | |
|---|---|---|---|---|
| Familiarity (*FAM*) | FAM→MOV | 0.293 | < .001 | *** |
| | FAM→ENE→MOV | 0.049 | < .001 | *** |
| | FAM→PLE→MOV | 0.036 | < .001 | *** |
| | FAM→INT→MOV | 0.020 | < .001 | *** |
| | FAM→REG→MOV | 0.006 | .005 | ** |
| | *FAM→⋯→MOV* | *0.403* | | |
| Pop. Music Affinity | Pop→PLE→MOV | 0.052 | < .001 | *** |
| | Pop→MOV | 0.041 | .070 | |
| | Pop→ENE→MOV | 0.039 | < .001 | *** |
| | Pop→INT→MOV | 0.026 | < .001 | *** |
| | Pop→REG→MOV | 0.001 | .520 | |
| | *Pop→⋯→MOV* | *0.159* | | |
| DSI Urge to Dance | DSI→MOV | 0.064 | .006 | ** |
| | DSI→ENE→MOV | 0.034 | < .001 | *** |
| | DSI→INT→MOV | 0.013 | < .001 | *** |
| | DSI→PLE→MOV | 0.012 | .032 | * |
| | DSI→REG→MOV | 0.002 | .184 | |
| | *DSI→⋯→MOV* | *0.126* | | |
| Complexity | Complexity→ENE→MOV | 0.042 | < .001 | *** |
| | Complexity→INT→MOV | 0.034 | < .001 | *** |
| | Complexity→REG→MOV | −0.027 | .005 | ** |
| | Complexity→MOV | −0.017 | .445 | |
| | Complexity→PLE→MOV | 0.011 | .021 | * |
| | *Complexity→⋯→MOV* | *0.042* | | |
| Female | Female→MOV | 0.013 | .533 | |
| | Female→PLE→MOV | −0.006 | .236 | |
| | Female→INT→MOV | 0.002 | .631 | |
| | Female→REG→MOV | 0.001 | .496 | |
| | Female→ENE→MOV | 0.000 | .958 | |
| | *Female→⋯→MOV* | *0.009* | | |
| MSI Training | MSI→MOV | 0.008 | .677 | |
| | MSI→PLE→MOV | −0.003 | .567 | |
| | MSI→REG→MOV | 0.002 | .147 | |
| | MSI→INT→MOV | 0.001 | .804 | |
| | MSI→ENE→MOV | 0.001 | .927 | |
| | *MSI→⋯→MOV* | *0.009* | | |
| Age | Age→MOV | −0.007 | .727 | |
| | Age→PLE→MOV | 0.007 | .166 | |
| | Age→REG→MOV | 0.004 | .014 | * |
| | Age→ENE→MOV | −0.003 | .687 | |
| | Age→INT→MOV | −0.002 | .535 | |
| | *Age→⋯→MOV* | *0.000* | | |

*Notes*: Significance probabilities are given for the entire pathways. Significance levels:.: *: $.010 \leq p < .050$; **: $.001 \leq p < .010$; ***: $p < .001$.

**Table 4. Effects mediated through *ENE*, *PLE*, *INT*, and *REG*.**

| Mediator | Regression path | Standardized Estimate (*b*) | *p* | |
|---|---|---|---|---|
| ENE | FAM →ENE→MOV | 0.049 | < .001 | *** |
|  | Complexity →ENE→MOV | 0.042 | < .001 | *** |
|  | Pop →ENE→MOV | 0.039 | < .001 | *** |
|  | DSI →ENE→MOV | 0.034 | < .001 | *** |
|  | Age →ENE→MOV | −0.003 | .687 | |
|  | MSI →ENE→MOV | 0.001 | .927 | |
|  | Female →ENE→MOV | 0.000 | .958 | |
|  | . . . →ENE→MOV | *0.161* | | |
| PLE | Pop →PLE→MOV | 0.052 | < .001 | *** |
|  | FAM →PLE→MOV | 0.036 | < .001 | *** |
|  | DSI →PLE→MOV | 0.012 | .032 | * |
|  | Complexity →PLE→MOV | 0.011 | .021 | * |
|  | Female →PLE→MOV | −0.006 | .236 | |
|  | MSI →PLE→MOV | −0.003 | .567 | |
|  | . . . →PLE→MOV | *0.109* | | |
| INT | Complexity →INT→MOV | 0.034 | < .001 | *** |
|  | Pop →INT→MOV | 0.026 | < .001 | *** |
|  | FAM →INT→MOV | 0.020 | < .001 | *** |
|  | DSI →INT→MOV | 0.013 | < .001 | *** |
|  | Age →INT→MOV | −0.002 | .535 | |
|  | Female →INT→MOV | 0.002 | .631 | |
|  | MSI →INT→MOV | 0.001 | .804 | |
|  | . . . →INT→MOV | *0.093* | | |
| REG | Complexity →REG→MOV | −0.027 | .005 | ** |
|  | FAM →REG→MOV | 0.006 | .005 | ** |
|  | Age →REG→MOV | 0.004 | .014 | * |
|  | DSI →REG→MOV | 0.002 | .184 | |
|  | MSI →REG→MOV | 0.002 | .147 | |
|  | Female →REG→MOV | 0.001 | .496 | |
|  | Pop →REG→MOV | 0.001 | .520 | |
|  | . . . →REG→MOV | *−0.010* | | |

*Notes*: Significance probabilities are given for the entire pathways. Significance levels:.:

*: $.010 \leq p < .050$

**: $.001 \leq p < .010$

***: $p < .001$.

recordings or GarageBand) and thus show a high degree of ecological validity. The drum patterns of the remaining 14 stimuli were composed by the experimenters with the intention to create patterns that have very low or very high values on the syncopation index. These stimuli expand the stimuli set's range on the index of syncopation.

We repeated the quadratic regression analysis using the Witek et al. [4] rating data relating to all 50 stimuli (see the comprehensive reanalysis of the 2014 data in S1 Text). This clearly confirmed the inverted-U hypothesis: a quadratic regression model significantly ($p < .001$) explained a substantial proportion of the variance between the urge to move ratings ($R^2 = .070$). When the 14 experimenter-composed patterns were dropped from the analysis, and the

quadratic regression model was fitted exclusively to the ratings in response to the 36 idiomatic patterns, the quadratic regression model was still significant ($p = .013$). However, it explained only a very small proportion of the variance in the urge to move ratings ($R^2 = .003$). The patterns that were sourced from original musical contexts received high urge to move ratings whereas the experimenter-composed patterns received low ratings (group difference: –0.955, $p < .001, R^2 = 013$). Because the experimenter-composed patterns are located at the extremes of the index of syncopation, it is mainly their ratings that bend the regression parabola downward for low and high syncopation values, thus forming the inverted-U-shaped function. If we compare same with same, i.e. only urge to move ratings in response to drum patterns taken from practical popular music contexts, the results are very similar across Witek et al. [4] and the current study.

Many replications of the inverted-U hypothesis show a similar configuration with musician-composed mid-complexity stimuli and experimenter-composed low or high complexity stimuli. Matthews et al. [9] and Cameron et al. [12] reused the entire Witek et al. stimulus set. Accordingly, their results can be interpreted along similar lines as Witek et al.'s. Cameron et al. ([12], supporting information) observed that the inverted-U function was weaker if the data of experimenter-composed high-complexity stimuli were dropped from the analysis.

In 2020, Witek et al. [15] used a subset of 15 stimuli from the 2014 Witek et al. [4] set: eight patterns were drummer-composed and seven were experimenter-composed. The mid-complexity group of stimuli exclusively featured drummer-composed patterns which obtained high mean urge to move ratings. The high-complexity group consisted of five experimenter-composed patterns, the low-complexity group mixed three drummer-composed with two-experimenter-composed patterns, and both groups received lower ratings compared to the mid-complexity group of stimuli.

The stimuli set from Matthews et al.'s 2019 [8] study featured two clave rhythms (son and rumba claves) and seven rhythms without specified origin. The two claves are idiomatic signature rhythms of dance-related Caribbean and Latin American music styles such as salsa, son and rumba. Both clave rhythms (plus one unspecified rhythm) were associated with medium complexity and obtained high urge to move ratings. The remaining unspecified rhythms formed the low and high complexity stimulus subsets, which were rated lower on the urge to move. In their 2022 study, Stupacher et al. [11] used a subset of three rhythms from Matthews et al. [8] to create their stimuli set: the son clave represented medium complexity whereas two unspecified rhythms represented low or high complexity. The son clave rhythm again obtained higher urge to move ratings compared to the two unspecified rhythms.

The medium complexity patterns of the 2024 Zalta et al. study [13] were originally played by musicians. Low and high complexity stimuli were then generated by the researchers through rhythmic manipulation: syncopation was removed to create low complexity stimuli, whereas syncopation was added in the case of the high complexity stimuli. The medium complexity stimuli were musician-composed and obtained high urge to move ratings. Low and high complexity stimuli resulted from researcher manipulation, and listeners rated them lower on the urge to move.

All studies presented in the paragraphs above confirmed the inverted-U hypothesis. And all of them used stimuli sets that predominantly use musician-composed musical patterns in the mid-complexity range and newly composed or rhythmically manipulated patterns in the low- and high-complexity range. One notable exception is the 2022 study by Spiech et al. [10], which exclusively presented experimenter-composed patterns and nevertheless confirmed the inverted-U hypothesis.

## Patterns matter

The results of Sioros et al.'s 2022 study [18] allow to better understand the role of musical syntax for the groove experience. Sioros et al. formed their stimulus set starting from ecologically valid popular music patterns (in funk or rock style, consisting of drums, bass, and guitar). They created new music examples by removing all syncopation algorithmically, thus creating deadpan versions of the patterns. They then introduced syncopation at random positions: the resulting music examples had a range between a few instances of syncopation (25% of the notes syncopated) and many instances (70% of the notes syncopated). They found that the original patterns obtained the greatest movement induction ratings, compared to the deadpan and randomly syncopated patterns. This implies that the ecologically valid idiomatic stimuli obtained greater urge to move ratings than the stimuli manipulated by the researchers (either removing syncopation or reintroducing at random locations).

Sioros et al. [18] came to the conclusion that the urge to move not only depended on the frequency or density of the syncopations, but to a great extent on the location of the syncopations within the musical structure. They summed up their results in the formula "patterns matter" (p. 503): it is relevant to the syntax of a musical structure, where in the bar and on which instrument syncopation appears. Sioros et al. observed that, in idiomatic patterns, syncopation rarely occurred in the cymbals (p. 514), that the backbeat in the snare drums is rarely syncopated (p. 512), that syncopated rhythms tended to form counter-rhythms to the dominant meter (cross-rhythmic or phase-shifted patterns, p. 515) and that pick-ups were often used to accentuate the beat (p. 516).

The current study's re-evaluation of earlier studies on the inverted-U hypothesis shows that the stimuli with musician-composed patterns received greater urge to move ratings than the stimuli with experimenter-composed or manipulated patterns. Listeners in an experiment cannot know how a pattern was created, but they are likely to notice whether a pattern is idiomatic and syntactically sound or not. Potentially, the low ratings of the experimenter-composed stimuli were not caused by their low or high complexity, but by the violation of syntactical rules. Music analytical frameworks as the one outlined by Sioros et al. [18] might be helpful in the future to study what it means to be idiomatic or syntactically sound for each type of musical pattern. Syntactical rules concerning popular music drum patterns could be derived based on a statistical analysis of large drum pattern corpora, such as the one presented by Hosken et al. [44].

## An alternative to the inverted-U hypothesis?

The discussion of stimuli sets and results across studies allows for the formulation of an alternative to the inverted-U hypothesis. In accordance with this study's results we claim that there is no systematic relationship between the rhythmic complexity and listeners' experience of groove–at least for isolated, idiomatic drum pattern stimuli that are within a complexity range which is typical for the respective repertoire. Informal support for this argument comes from the dance music repertoire itself: countless people dance every day to rhythmically simple (e.g. eurodisco, eurodance) or complex (e.g. Cuban salsa, Brazilian samba, Macedonian pušteno) music. This indicates that the complexity of music may not be very important to listeners' motivation to move in general. Rather, it may be more important whether patterns are syntactically sound within a style, whether listeners are familiar with the musical style and thus capable of decoding the metrical regularities underlying the rhythmic structure, whether listeners enjoy the music and whether it is traditionally used for dancing. All this may be much more relevant to listeners' motivation to move than the mere level of complexity. Past confirmations of the inverted-U hypotheses potentially resulted from the confounding between measures of complexity (such as the index of syncopation) and the syntactical soundness of the musical patterns.

## Structural equation model (SEM)

The SEM implementation of the groove model, allows to widen the perspective beyond complexity and to simultaneously consider the importance of personal background factors to the experience of groove. It is sensible to compare this study's SEM to the model presented in the first study that used the groove model SEM implementation [33] in order to study which aspects of the model are stable, and which are variable. Preprints of two additional studies using a similar SEM modelling approach have recently been published [45, 46] and provide further contextualisation.

**General properties.** This study's SEM had a very good fit, just like the SEM of [33], yet the two studies differ in their capacity to explain the variance of the *MOV* variable. This study's SEM explains a considerable proportion of the variance in *MOV* ($R^2$ = .482), yet it should be noted that the earlier study's SEM ([33], p. 297) was far more successful in explaining the variance in *MOV* ($R^2$ = .737). This difference can mainly be attributed to one reason: In the earlier SEM model, the stimuli explain a large proportion of the variance in the *MOV* latent variable ($R^2$ = .240; [33], p. 296). In the current study, only one musical predictor variable (perceived stimulus complexity) entered the model; the stimuli were not included as predictors. The difference in the $R^2$ effect size may to a substantial degree depend on the fact that, in this study's SEM, the musical properties have very little relevance as predictors for *MOV*.

**Correlations between latent variables.** Senn et al. ([33], p. 295) reported very high positive correlations between *INT*, *PLE*, and *ENE*, ranging from $r$ = .664 (*ENE*, *INT*) to $r$ = .758 (*PLE*, *INT*). These strong correlations might at least partly be due to the fact that the questionnaire items relating to the four mediators were all collected in the same trials on the same page of the questionnaire, and only *MOV* was collected separately. In the current study, data regarding each latent variable were collected in separate trials. As expected, the correlations between *INT*, *PLE*, and *ENE* were less strong (**Fig 4**), ranging from $r$ = .482 (*INT*, *ENE*) to $r$ = .543 (*INT*, *PLE*; see also Table 8 in S1 Appendix, Standardized Estimate). Collecting the data for the latent variables in separate trials reduced spurious positive correlations and resulted in greater independence of the latent variables. This finding may also offer a nuance with respect to the very high positive correlations of the *MOV* and *PLE* latent variables found in the studies presenting different language versions of the Experience of Groove Questionnaire in English ($r$ = .80; [37], p. 56) and German ($r$ = .90; [42], p. 93). A Japanese version of the questionnaire is currently in preparation and also shows high positive correlations between *MOV* and *PLE* ($r$ = .77). In these studies, *MOV* and *PLE* ratings were collected in the same trials, which partly explains the high correlations between the two scales. In the current study, the ratings for *MOV* and *PLE* were collected in different trials, and the positive correlation between the scales was medium-sized ($r$ = .510).

**Familiarity.** Familiarity (*FAM*) is a strong and highly significant predictor for all latent variables (**Fig 4**). People who stated to be familiar with the music, also tended to give high ratings of *REG*, *INT*, *PLE*, and *ENE*. The high coefficient for the effect of *MOV* needs to be interpreted with a grain of salt, however, because *FAM* and *MOV* were collected in the same trials, which likely exaggerated the association between the two scales.

**Popular music affinity, DSI, and MSI.** All stimuli were reconstructions of drum patterns from Western popular music. The *popular music affinity* variable measures listeners' attitude towards these styles. It turned out that popular music lovers tended to find the stimuli more interesting, pleasurable, and energizing than people who don't like Western popular music. This underscores the relevance of music preference for the groove experience [17, 47, 48].

Similarly, people who generally feel compelled to dance (DSI, urge to dance scale) also felt more interest, pleasure, energy and urge to move in response to this study's stimuli, compared

to people who in general are not eager to dance. This result is in agreement with Witek et al. [4], but it contrasts with O'Connell et al. [49], where no effect of the urge to dance scale on listeners' sensitivity to groove was found.

Listeners' musical training affected the urge to move ratings in some previous studies [8, 16, 47], but did not have a significant effect on the urge to move [4] or any of the latent variables [33] in others. In the context of this study, musical training was not relevant to the groove experience.

## Limitations

The inverted-U hypothesis ($H_1$) was not confirmed by this study. There are a few limiting factors in the design of the study that might have contributed to this:

- The forty drum pattern stimuli are a subset of 250 drum patterns that were explicitly selected to represent the work of highly renowned drummers and to include iconic patterns ([17], pp. 5–7). This might be considered a selection bias: the selected passages in the originally recorded sources all presented high quality drumming, both in terms of the recorded patterns and of the performance skills of the original performers. It is difficult to assess, to which degree the MIDI reconstructions conserved the subtle properties of the original performance. Yet, the fact that every reconstructed stimulus had a high-quality original source, regardless of the complexity of the pattern, might have overridden a potential inverted-U relationship between complexity and the urge to move in the general population of popular music drum patterns.

- The stimuli only represent a range of complexity that is idiomatic for mainstream pop, funk, or rock music in 4/4 common time. Different meters and more experimental genres are not represented in the sample. Accordingly, the upper limit of the complexity range does not go into extremes. Maybe, a decline of urge to move ratings could have been observed for stimuli with high complexity due to uncommon time signatures (on the topic of uncommon meters and groove, see the preprint by Spiech et al. [50]).

- One further limitation is innate to the groove model and its SEM implementation: the collection of data in five different experimental blocks decoupled the five latent variables of the model, which shows in the reduced correlations between the latent variables in the current study, compared to [33]. On the negative side, this procedure made the experiment long, repetitive, and potentially boring for the participants. Each participant only judged ten stimuli, which covered a wide complexity range. Results would have been more informative, if every participant rated all forty drum patterns on all latent variables. But in this case the survey would have taken too much time to fill. We hope that the benefits of the model (a more holistic approach to groove research) makes these sacrifices worthwhile.

- This study (as most studies in the field) represents an exercise in reductionism: we work with drum patterns that may be heard alone occasionally in daily music listening, but that usually are a part of a much denser full band texture. We drew these passages from three major popular music styles (rock, pop, funk), which only represent a fraction of music that humans use for dancing. The listening situation is reduced to an audio-only modality where participants experience short excerpts of music at their computer or tablet. In real-life situations, experiencing music often includes different modes of perception, it may include actual body movement, and listening usually takes much longer than the few seconds of duration provided by this study's stimuli. Duman et al. [48] rightly point to the multi-faceted nature of the groove experience; and the groove model accounts for many facets at least in theory

(see [32, 33]). More ecologically valid listening situations have been implemented in past research (such as the realistic concert or clubbing situations created by Swarbrick et al. [51] or Cameron et al. [52]). However, behavioural music psychological research will always have to find an equilibrium between experimental control and the naturalness of the listening situation.

## Conclusions

This study was designed to test the hypothesis that music listeners' urge to move in response to popular music drum patterns is an inverted-U function of the drum patterns' complexity ($H_1$). The hypothesis was not supported by the data. Rather, the data suggests that stimulus complexity and the urge to move are unrelated. This result is at odds with the findings of a number of studies that confirmed an inverted-U relationship between syncopation and the urge to move, assuming that syncopation is a rhythmic device that adds complexity. The re-analysis of the 2014 Witek et al. [4] data shed some light on this matter: the relationship between syncopation and the urge to move showed a strong inverted-U shape if all of Witek et al.'s stimuli were included in the analysis. Yet, the inverted-U relationship was weak when only idiomatic drummer-composed drum pattern stimuli were used and the experimenter-composed patterns were excluded. Thus, if we compare same with same (i.e. only responses to idiomatic drum patterns), the results were quite similar across the two studies. Many studies that confirmed the inverted-U hypothesis used potentially unidiomatic musical patterns for low or high complexity stimuli. This may have led to confounding between complexity and syntactic soundness as predictors of the urge to move ratings. Accordingly, the importance of the stimuli preparation for listening experiments cannot be overstated. Stimuli need to be plausible representants of an existing musical repertoire, otherwise the study results may (in a worst-case scenario) only reflect the difference between idiomatic and unidiomatic music examples. As of today, it is unclear to what extent the present study's findings on the inverted-U hypothesis can be generalised. Ideally, authors of all existing studies about the relationship between syncopation (or complexity) and the groove experience join forces in order to re-analyze the combined data in a meta-study and offer a joint perspective on the inverted-U hypothesis.

Hypotheses $H_2$ and $H_3$ translated Stupacher et al.'s [29] "sweet-spot" idea into the domain of the groove model. These hypotheses claimed that, in drum pattern stimuli of different complexity, two contrasting principles were at work: simple patterns invite an urge to move because it is easy to synchronize body movement with the music; and complex patterns motivate movement, because they keep listeners interested. The experiment provided strong evidence in support of both of these mechanisms. Yet, no optimal "sweet spot" at medium complexity was found that maximizes the urge to move. Instead, the negative effect through *REG* and the positive effect through *INT* cancelled each other out.

In this study, the SEM implementation of the psychological groove model proved to be a useful framework for data analysis: it made it possible to formulate and test two of the hypotheses that involved mediation. It allowed to simultaneously consider musical properties and personal background factors (such as listeners' familiarity with the music or their preference for popular music styles). The model's greatest strength, potentially, is the use of a well-defined model structure and standardized regression coefficients. They allow to compare effects across studies and they offer a frame of reference that provides both stability and flexibility: the stability depends on the fact that the standardized regression coefficients will have the same interpretation across different study designs. The flexibility derives from the possibility that stimuli, exogenous variables, and mediators can be changed and/or reorganized to investigate different modelling ideas and repertoires.

## Supporting information

**S1 Appendix. Appendix.**
(PDF)

**S1 Text. Re-analysis of the Witek et al. (2014) [4] data.**
(PDF)

## Author Contributions

**Conceptualization:** Olivier Senn, Florian Hoesl, Toni Amadeus Bechtold.

**Data curation:** Olivier Senn, Florian Hoesl, Maria Witek.

**Formal analysis:** Olivier Senn.

**Funding acquisition:** Olivier Senn.

**Investigation:** Olivier Senn.

**Methodology:** Olivier Senn, Florian Hoesl, Toni Amadeus Bechtold.

**Resources:** Florian Hoesl, Toni Amadeus Bechtold, Maria Witek.

**Visualization:** Olivier Senn.

**Writing – original draft:** Olivier Senn.

**Writing – review & editing:** Olivier Senn, Florian Hoesl, Toni Amadeus Bechtold, Lorenz Kilchenmann, Rafael Jerjen, Maria Witek.

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
