## [Decision Letter · Decision Letter 0]

21 Jun 2024

PONE-D-24-17648Null effect of perceived drum pattern complexity on the experience of groovePLOS ONE

Dear Dr. Senn,

Thank you for submitting your manuscript to PLOS ONE. Your manuscript, referenced above, has now been reviewed by experts in the field. After careful consideration, we feel that it has merit but does not fully meet PLOS ONE’s publication criteria as it currently stands. The reviewers have made some suggestions, which the Editor feels would improve your manuscript. We encourage you to consider these comments and make an appropriate revision of your manuscript. Therefore, we invite you to submit a revised version of the manuscript that addresses the points raised during the review process. The comments of the reviewers are included below in order for you to understand the basis for our decision, and we hope that their thoughtful comments will help you in your revision.

 Please submit your revised manuscript by Aug 05 2024 11:59PM. If you will need more time than this to complete your revisions, please reply to this message or contact the journal office at plosone@plos.org. Please include the following items when submitting your revised manuscript:A rebuttal letter that responds to each point raised by the academic editor and reviewer(s). You should upload this letter as a separate file labeled 'Response to Reviewers'.A marked-up copy of your manuscript that highlights changes made to the original version. You should upload this as a separate file labeled 'Revised Manuscript with Track Changes'.An unmarked version of your revised paper without tracked changes. You should upload this as a separate file labeled 'Manuscript'.

We look forward to receiving your revised manuscript.

Kind regards,

Phakkharawat Sittiprapaporn, Ph.D.

Academic Editor

PLOS ONE

Journal Requirements: 

 "Swiss National Science Foundation (Grant No. 100016 192398 to Olivier Senn)."

5. Thank you for uploading your study's underlying data set. Unfortunately, the repository you have noted in your Data Availability statement does not qualify as an acceptable data repository according to PLOS's standards.

Reviewers' comments:

**Comments to the Author**

1. Is the manuscript technically sound, and do the data support the conclusions?

Reviewer #1: Partly

Reviewer #2: Yes

Reviewer #3: Yes

Reviewer #4: Partly

2. Has the statistical analysis been performed appropriately and rigorously? 

Reviewer #1: I Don't Know

Reviewer #2: Yes

Reviewer #3: Yes

Reviewer #4: Yes

3. Have the authors made all data underlying the findings in their manuscript fully available?

Reviewer #1: Yes

Reviewer #2: Yes

Reviewer #3: Yes

Reviewer #4: Yes

4. Is the manuscript presented in an intelligible fashion and written in standard English?

Reviewer #1: Yes

Reviewer #2: Yes

Reviewer #3: Yes

Reviewer #4: Yes

5. Review Comments to the Author

Reviewer #1: 

SUMMARY

The submission "Null effect of perceived drum pattern complexity on the experience of groove" contributes to the field of psychological groove studies that investigate the question of what it is that triggers a pleasant urge to move (PLUMM, after Bechtold, Curry, and Witek 2024, PlosOne) when listening to music. The field is still young and small but has been flourishing for around fifteen years; the focus of research is on the temporal organization of music (timing, rhythm, meter, and ensemble synchronization). The most prominent hypothesis in the field is that music must be of moderate temporal complexity to convey maximum PLUMM. This has been expressed as the inverted U-shape or sweet spot hypothesis for moderate temporal complexity predicting PLUMM: assuming that a relatively low degree of temporal complexity involves a lack of surprise/interest, which leads to boredom resulting from over-predictability and that a relatively high degree of temporal complexity involves an excess of surprise which leads to disorientation resulting from under-predictability. It is thus predicted that both very low and very high levels of complexity are associated with suboptimal levels of PLUMM; the apex of the U is explained as the sweet spot between over- and underpredictability. This U-shape/sweet spot hypothesis has been tested (Witek et al. 2014, Soros et al. 2014) and replicated in several studies (e.g., Stupacher et al. 2022) over the past decade with good success, although some replications have also failed.

The present submission presents an extended replication study, which re-tests the U-shape hypothesis with new stimuli (a selection of popular music drum patterns tested for perceived complexity) with well-developed response measures (perceived urge to move, temporal regularity, rhythmic interest, listening pleasure, and energetic arousal) and statistical approaches. The study fails to replicate the U-shape; this is surprising relative to the state of research in the field. The study then finds a two-fold mediating mechanism that relates complexity and PLUMM: (i) low-complexity stimuli afford regularity which indirectly enhances PLUMM without also involving the negatively correlated aspect of boredom/lack of interest (which is often assumed to come with low-complexity stimuli); (b) high-complexity stimuli afford rhythmic interest which indirectly enhances PLUMM without also involving the negatively correlated aspect of disorientation/lack of regularity (which often is assumed to come with high-complexity stimuli). However, the study also fails to replicate the sweet-spot formulation of the U-shape hypothesis: the two mediating tendencies do not generate a sweet-spot of particularly strong enhancement of PLUMM in the vicinity of the peak of the U.

GENERAL ASSESSMENT

The study is relevant and interesting. It is very well crafted and presented. I don’t take an issue with the materials, methods, analyses, and findings. The stimuli are based on a plausible and empirically justified selection of musical materials, and they are synthetically re-constructed with great care, allowing for an admirable combination of ecological validity and control of variables. The measures taken from participants’ self-assessing their listening experience make much sense, and are formally justified and interpretable from a previously published psychological PLUMM-centered model of groove (Sen et al 2019). In general, the study design is incrementally building on, critically self-questioning, and contributing further to earlier work by the first author (Senn), the last author (Witek), and their colleagues. This is best practice; it allows for exceptional conceptual clarity, methodological transparency, and analytical reliability.

The main purpose of this review is to encourage the authors to reconsider the interpretation of their findings in the discussion and conclusion sections, which appear suboptimally structured, partly unbalanced, and partly too narrow.

SECTION “POSSIBLE REASONS WHY THE INVERTED-U HYPOTHESIS WAS NOT CONFIRMED”

In this section, the authors announce to discuss “Why was the inverted-U hypothesis not confirmed in this study?” Unfortunately, the biggest portion of this section does not answer this question but rather discusses why earlier studies might have found it in contrast to the present one. This discussion of Witek et al. 2014 is relevant and productive but should be dealt with only after answering the original one: discussing why the U-shape was not found in the current study.

A smaller argument provided by the authors is that the stimuli might not be representative but too good in that their representing exceptionally high degrees of musiciansship might involve a kind of ceiling effect that masks the U-shape. I believe this is a relatively minor point and relatively unlikely explanation of the findings. Moreover, it is redundantly discussed also in the sections on Limitations. I suggest combining those two discussions; the section on Limitations appears to be the better place.

Curiously, that’s all the authors say about why they think they did not find the U-shape. That’s virtually nothing. It reads almost as if they wanted to explain away their main finding rather than discuss it. In particular, the most obvious possible explanation of the findings is not discussed: The possibility that the U-shape does not show up in the studied data because temporal complexity might not be a good predictor of PLUMM, either because PLUMM is relatively independent of temporal complexity or because the effect is so weak that it can easily be overridden or masked by other, more important factors.

The confirmation of two secondary hypotheses tested in the study can shed some indirect further light on the interpretation of the main finding. The study confirmed, on the one hand, a direct negative effect of complexity on regularity and a related indirect negative effect of complexity on PLUMM. On the other hand, the study found a direct positive effect of complexity on rhythmic interest and a related indirect positive effect on PLUMM. In sum, complexity seems to have negative and positive indirect effects on PLUMM. In the conclusion, the authors interpret this combination of main and secondary findings as follows: “The study found a psychological mechanism [the two divergent mediating effects] that potentially explains how the inverted-U hypothesis works, but it could not confirm the hypothesis itself.” This appears one-sided and is a bit perplexing; it reads as if the authors were mainly interested in defending the U-shape hypothesis rather than questioning it. As far as I understood the study, its findings are consistent with an at least equally plausible reading: complexity might not be a good predictor of PLUMM, and the U-curve might not even exist as a general psychological principle. Please correct me if I am wrong; I am happy to learn how to understand the findings better. However, if I am not completely wrong, the possible explanation I outlined should be made explicit and put at the center of the discussion.

I here take the liberty to address two insights from neighboring research fields that appear consistent with the assumption of flexibility and contingency in the complexity-to-PLUMM relation. First, my intuition as a comparative ethnomusicologist suggests that there are both rhythmically simple and rhythmically complex musical practices that equally make millions of people dance. For instance, Eurodisco and Eurodance can be assumed to rank low on syncopation and perceived complexity indexes, whereas Samba and Salsa would rank high. Yet, these genres are all known to be very successful in providing PLUMM and actually making people dance. Second, cross-cultural studies of rhythm/meter perception have shown that familiarity can override complexity to a surprisingly large extent (e.g., Hannon et al. 2017). These two arguments do not speak against the possible existence of a general psychological mechanism that relates complexity to PLUMM. However, they seem to suggest that such a mechanism, if existing, can be overridden or masked with relative ease, e.g., by the processes of enculturation and familiarization with specific musical genres, processes that all humans inevitably go through.

Section “LIMITATIONS”

The authors mention that a possible limitation of the study is that the stimuli were modeled after performances by highly competent musicians. In the same vein, they stated earlier in the discussion that “Great drummers can play a simple pattern in such a way that it is still interesting to the listener, and they ca [sic] present a complex pattern such that listeners still feel the regularity.” I am skeptical about this representing a limitation. First, I wonder how much of the exceptional skills of the musicians would show up in the stimuli. For instance, the synthesized stimuli do not represent their sounds and micro-dynamics. Furthermore, most aspects of skilled groove performance in the original recordings will have to do with their playing in a polyphonic ensemble context and interacting with their ensemble partners, aspects not represented in the stimuli. Second, alluding to exceptionally skilled individuals contradicts the author's own claim in this and other studies, to have constructed the stimuli as “a representative sample of highly competent musicians within the genres of … “ (Senn et al. 2018). In summary, I would argue that this limitation is unlikely to have a considerable effect. Assuming that the selected stimuli are indeed representative of certain musical genres, my reading of the findings in the current study is that these musical genre traditions, as represented by competent musicians, have found ways to generate PLUMM with both high and low degrees of rhythmic complexity. While explicating limitations is, of course, laudable, I would recommend ranking this one relatively low.

The authors then mention as a limitation that the stimuli cover a range of complexity that is limited to the range that is typical of the represented genres. Again, I do not think this is a limitation of the study but rather adds to its ecological validity. They suggest that a solution would be to add artificial stimuli with extreme complexity (contradicting their original aim to construct a stimulus set that is representative). This is exactly what Witek and colleagues (2014) did, which the authors productively deconstructed as a somewhat arbitrary approach that introduces a degree of unreliability when it comes to interpreting the results. Artificial stimuli involving extreme degrees of variation, of course, can be very useful in psychological experiments, but mixing these with realistic stimuli allegedly representing specific social/cultural traditions appears odd.

Beyond the limitations mentioned by the authors, I think there are more serious ones that might deserve to be mentioned.

First, the music genres used as materials for study are polyphonic and polyrhythmic (polyphony derived from the coordination of contrasting rhythms in multipart ensembles). It has been prominently argued in humanities-based groove research (e.g., works by Anne Danielsen) that such polyrhythm is a core component of the groove experience. By contrast, psychological groove research relies on monophonic representations of this polyphonic groove music. In particular, the auditory stimuli and measures used to quantify complexity/syncopation are based on the drum set to the exclusion of the bass, rhythm guitar, and keyboard parts and their obviously relevant interactions; not to speak of song and dance, which can be safely assumed to strongly contribute to the groove experience in groove music, too. This reduction to the drum set has a degree of intuitive plausibility and certainly is logistically convenient. Yet it obviously also represents a limitation, the empirical justification of which might be a productive route for future research: e.g., experiments measuring how much of the groove rating for a set of full musical excerpts can be explained by its drum-part alone should be interesting.

Second, the selection of three genres used for stimuli construction (rock, funk, and mainstream pop) is narrow. Many hugely successful genres of groove-based, including some of the globally most widely selling ones (e.g., hip-hop, EDM, techno, Latin trap, Afrobeats, k-pop, etc.), are not represented for unspecified reasons. The selected range of genres thus appears as a convenience sample, which is a limitation. One potentially advantageous implication of using a larger diversity of genres in future research would be to fulfill the desideratum of increasing the overall variation (perceptual space) on the complexity dimension without using artificial examples. If realized in future research, such diversification of genres in the stimuli would perhaps also increase the motivation to correspondingly increase the cultural/genre-specific diversification in the participant samples. One argument favoring acknowledging genre-specificity and genre diversity more broadly than hitherto is the Sioros et al. 2022 paper mentioned in the introduction. This study finds that neutral mathematical measures of syncopations do not correctly predict groove ratings because groove depends on genre-specific patterns, which add subjective weightings to locally specific types of syncopation. Such patterns likely are genre-specific in a double-sense, relating to both genre-specific musical repertoires and genre-specific familiarities and expectations on the perceivers’ side.

MORE LIMITATIONS

Alongside the above limitations relating to the details of the auditory stimuli used in the present study, it might also be worth going one level of abstraction higher and reflecting on the concept of psychological groove research a bit more generally. The following limitations concern the present study and the field at large.

First, PLUMM is excusively tested in auditory stimuli. In contrast, it can be safely assumed that groove is a multisensory experience. In particular, the visual modality will be highly relevant for conveying musical properties to observers (cf., the relevance of movement in music video clips), while the somatosensory modalities (especially proprioceptive, vibrotactile, and vestibular) will be relevant for first-person experiences.

Second, it is obvious that groove is an experience that does not stop at the urge to move but also has actual movement as a core component. This has been argued, for instance, by the submission’s last author (Witek 2017). Movement is manifest both in the music-making itself (in the narrower sense) and in the co-constitutive performance modalities of dancing, clapping, head-bobbing, entraining one’s locomotion, etc. (the musicking in the broader sense). From this perspective, it appears unintuitive that groove research does not consider actual movement to any considerable degree. At the method level, measures for PLUMM mostly rely on participants’ self-assessment of the listening experience. In contrast, the urge to move is hardly ever tested with actual movement used as a measure. Relevant methodological approaches to link acoustic musical properties to movement response have been developed, for instance, by the Toiviainen (Jyväskylä) and Leman (Gent) labs (see, for instance, work by first authors Birgitta Burger and Luiz Naveda, respectively). At the level of theory, the model used in the current submission (Senn et al 2019) focuses on the urge to move to the relative exclusion of actual movement and its social implications, which stands in contrast to the movement-centered understanding of Witek 2017 and the social aspects of groove (shared state of mind, interaction) emphasized in humanities-based groove-research; the latter social aspects of groove also may well be related to the vast field of social psychology research into the prosocial effects of rhythmic entrainment. Senn’s narrower model has its justification, for instance, allowing the formulation of testable hypotheses, but it certainly represents a relatively narrow view, one that invites being discussed in the context of what a broader view, including neighboring fields (e.g., interpersonal entrainment) might involve.

Admittedly these latter limitations concerning the multimodal and multisensory, movement-related, and social and interaction-related aspects of groove are very hard to tackle when it comes to empirical research. It might be worthwhile considering them in the context of the current submission nonetheless. This submission is special: two eminent, otherwise independent lab leaders and lead authors (with collaborators) have come to work together, for the first time, it seems; they together created an excellent empirical study that falsified in an excellently developed and relevant data-set the core hypothesis in their common field of research, questioning their own earlier research. In my view, this is a great context to take a step back and discuss also the more general issues and possible futures of the field. However, I see this as facultative. The submission would work well also without responding to what I addressed under the heading of MORE LIMITATIONS.

SMALLER POINTS

Abstract:

“Danceablility” is used several times, apparently synonymous for groove. Is there a special reason to single out the dimension of danceability in the abstract while the main text consistently relates to the higher-level concept of groove?

L231

the music wit (typo)

L233

to what extent feel (grammar)

L477

and they ca present (typo)

L526

Predicotr (typo)

Reviewer #2: 

In this study, the authors test the inverse U-shaped hypothesis linking rhythmic music complexity with the urge to move, known as Groove ratings, in humans. Typically, the level of syncopation is used as a proxy for rhythmic complexity, and most previous articles have investigated the effect of syncopation on Groove ratings. 

Here, the authors test the inverse U-shaped hypothesis using forty idiomatic popular music drum pattern stimuli associated with perceptual complexity measures obtained in a previous study involving 179 participants. They do not find the inverse U-shaped relationship observed in previous studies using this perceived complexity measure.

Testing different metrics of rhythmic complexity, rather than relying solely on the level of syncopation, seems important and useful for understanding the emergence of Groove behavior when humans listen to music. However, I currently have major and minor concerns about the methodological aspects of the study and the tools used to assess their hypotheses.

Please see below my detailed comments and suggestions:

Major revisions

1. The authors did not test (or show) the relationship between syncopation and MOV level for their stimuli. It would be interesting to do this to see if the authors replicate the findings in the literature using an equivalent metric. Moreover, comparing the perceived complexity with the level of syncopation would help to understand why, even if these two metrics are well correlated, the authors do not replicate results shown by many different studies. Additionally, I wonder if the range of syncopation in the dataset is too narrow to reveal a clear inverse U-shaped curve.

2. The empirical perceived complexity measure that the authors used instead of the level of syncopation to test the inverse U-shaped hypothesis is not clearly defined in the manuscript (even though the authors relate it to the seminal paper of this measure). It would greatly enhance the readability of the article to better explain this measure and how it diverges from the level of syncopation in the methods section.

3. Perceived complexity seems to be a subjective measure of musical stimuli. Could the absence of an inverse U-shaped relationship between this metric and MOV be due to potential covariations? Even if the index of syncopation and perceived complexity are correlated (line 60), the inverse relationship found between MOV and the index of syncopation could arise from other effects.

Minor revisions

1. Line 231: “Participants’ enjoyment while listening to the music wit three items and range…” Please correct this typographical error.

2. Line 526: “In the current study, only one musical predicotr variable (perceived complexity) entered the model…”. Please correct this typographical error.

3. Line 124: the acronym SEM is introduced but only explained later in the caption of Figure 3 (line 384) and again in the discussion section (line 514). To enhance readability, the authors should explain what SEM means and its relevance before first using it.

Reviewer #3: 

First of all, I would like to thank the authors for their excellent paper, which I have read with great interest and pleasure.

This paper examines the relationship between groove ratings and complexity, where groove is understood as the pleasurable urge to move to music (PLUMM). In particular, the paper looks at the inverted U-shaped relationship previously found between groove and PLUMM (Witek et al. 2014), which prompted the predictive coding modelling of groove. At the same time, the paper attempts to explain the experimental data in terms of a psychological model of groove that consists of a number of components that influence our urge to move. Most relevant to this paper are 1) stimulus complexity, 2) interest (INT), 3) pleasure (PLE), 4) perceived regularity (REG), 5) energetic arousal (ENE), and 6) personal background. The model proposes that complexity affects our urge to move indirectly through the above components.

A dataset of 40 pop drum loops, originally performed by renowned and famous drummers and reconstructed/produced from MIDI, is used in a listening experiment in which ratings were collected on urge to move as well as other scales determining all the above components of the psychological model of groove. The loops were rated by listeners in a previous experiment for their perceived complexity.

The paper did not confirm the inverted U relationship between complexity and urge to move for these music examples. However, it did confirm the indirect effect of complexity on our urge to move within the psychological model of groove and reported regression coefficients for all relevant pathways.

In addition, the paper re-analyses the data from Witek et al. 2024, where the inverted U relationship was first reported, and shows that this inverted U relationship was primarily driven by the use of both commercial drum patterns and experimenter-designed patterns that were rated as "less groovy". The designed patterns were either very simple or "extremely" complex, while the commercial patterns were moderately complex, resulting in the inverted U-shape. When the data collected was analysed using only the commercial patterns, the effect of complexity on the urge to move virtually disappeared.

The paper is well written, the experiment carefully designed and the analysis thorough. The rejection of the inverted U hypothesis is a particularly important finding, given how influential it has been in driving the predictive coding interpretation of the groove experience.

I therefore recommend that the paper be accepted for publication.

Some more specific questions/comments:

1) In lines 494-500, the inverted U-shape of Witel et al. is attributed to the patterns designed specifically for that experiment, which were rated lower on "urge to move" compared to other commercial drum patterns. This result is very similar to that reported by Sioros et al 2022, where the original syncopated patterns of commercial songs were rated higher than algorithmically syncopated patterns of the same songs. Sioros et al. put forward a number of hypotheses as to what makes the original patterns groovier than the algorithmic ones. One hypothesis is that the original syncopated patterns more often form "cross rhythms" (e.g. 3 over 4). Another hypothesis is that the original patterns feature more pickups.

While I do not necessarily ask for any further analysis, I wonder if some analysis regarding cross-rhythms and pickups could also be done in the Witek et al. (2014) dataset.

2) I find that lines 8-10 of the abstract are not very clear as to what the novel contribution of this paper is. As mentioned in the paper, this is not the first study to try to replicate the inverted U shape. But the wording seems to me a bit ambiguous as to what makes this study the first!

3) A reference to a definition of syncopation may be needed in lines 80-82.

4) Lines 470-480 suggest that skilled drummers can perform simple patterns in interesting ways and complex patterns in a way that still sounds regular, and I agree with this claim. But doesn't this mean that other features of music are actually more important than complexity for our urge to move? Whether these features also contribute to complexity may be "accidental".

5) Lines 5-8: Syncopation is not linearly related to complexity, even though complexity and syncopation have sometimes been used almost interchangeably in the literature. For example, the highly syncopated patterns of Witek et al. 2014 are complex, but the highly syncopated patterns of Sioros et al. 2014 are simply isochronous patterns with a phase difference to a metronome and probably not perceived as complex. I do not disagree with the specific lines in the abstract, but perhaps this should be further clarified in the text (see for example lines 28-29).

6) line 510-512: This is an excellent result! I would even remove the word "every-day". The experience of groove is genre specific, even if there may be some universal principles behind it.

In line 477, there is a typo (ca  can).

Reviewer #4: 

Summary of the study

This paper investigates three hypotheses: whether the strongest groove is achieved by drum patterns with medium complexity (inverted U-shaped relationship between groove and rhythmic complexity) (H1), whether rhythmic complexity negatively affects temporal regularity (H2), and whether rhythmic complexity positively affects the listener's interest in drum patterns (H3). Instead of employing calculated syncopation as an index of rhythmic complexity, the authors used perceived complexity. Using an SEM approach, they confirmed H2 and H3, which corroborated the "sweet spot" idea by Stupacher et al. (2022). They argued that there is a sweet spot between predictability and surprise; the former elicits the groove experience by allowing listeners to synchronize with the music easily, and the latter motivates listeners to move by keeping them interested. However, H1 was not corroborated by their results. After re-analyzing the data of Witek et al. (2014), which showed an inverted U-shaped relationship, they found similar results when analyzing only the idiomatic patterns, excluding the experimenter-composed patterns. They concluded that the results of Witek et al. (2014) (inverted U-shaped relationship) depended highly on the experimenter-composed patterns, while there is no such relationship for idiomatic patterns.

Major comments

Firstly, I like this study very much. The study has clear hypotheses and appropriate methods of analyses. Additionally, the results are interesting and advance our understanding of the relationship between the groove experience and rhythmic complexity. However, there are several points that need to be clarified.

1. Combining the results of this study and the re-analysis of the data from Witek et al. (2014), the authors concluded that "stimuli need to be plausible representatives of an existing musical repertoire; otherwise, the study results may (in a worst-case scenario) only reflect the difference between idiomatic and un-idiomatic music examples." I partially agree with this but cannot completely concur. In my opinion, if we want to investigate the relationship between the groove experience and rhythmic complexity in general, we don't necessarily need to restrict the stimuli to idiomatic, or commonly heard, examples. Although not idiomatic, there indeed exists music with highly complex rhythms (e.g., Iannis Xenakis, Mars Volta, etc.). Therefore, my suggestion for interpreting the results is that there is an inverted U-shaped relationship when examining the general relationship between groove experience and rhythmic complexity (as shown by Witek et al., 2014). However, when focusing on idiomatic rhythms (or music), this relationship does not appear. This is likely because well-known drummers can choose the right complexity (not too simple or too complex, reflected in the relatively narrow complexity range) and have the ability to play simple rhythms interestingly and complex rhythms with high regularity (as the authors discuss, which I fully agree with).

2. If rhythmic complexity has a negative effect on temporal regularity and a positive effect on time-related interest, it suggests that there is an optimal complexity (i.e., an inverted-U relationship). You mentioned this in the limitations, but could you explain more about this result? You stated, "the study found a psychological mechanism that potentially explains how the inverted-U shaped hypothesis works, but it could not confirm the hypothesis itself." Why do you think this happened? I was very interested in the results of SEM supporting the optimal "sweet spot" idea, so I would like to know your opinion on why this occurred.

3. I have a concern that the perceived complexity used in this study does not come from the participants of this study (as you mentioned in the limitations). Is there any reason you did not collect the perceived complexity data from the participants this time? Are the results of the previous study regarding perceived complexity rigorous?

Minor comments

1. line 95

It is better to cite a reference such as Vuust et al. (2022).

2. line 105

Do you have a citation for the following statement?

"Synchronized body movement (resp. dancing) allows a listener to process rhythmic complexity more easily"

3. line 129

"a series psychometric scales" → "a series of psychometric scales"?

4. line 231

"wit" → "with"

5. line 270

Does "s" mean "SD" or "SE"?

6. line 300

Why did you always present the MOV & FAM block first?

7. line 362

What is the range of the calculated syncopation in this study? The readers (including myself) would likely be interested in the difference of the range between this study and Witek et al. (2014).

8. line 398

Is it possible to investigate non-linear relationship (e.g., quadratic relationship) using the SEM approach? You don't have to try this for this study, but I was just curious if the time-related interest linearly increases as the complexity becomes higher. There might be an inverted-U relationship between the two.

9. line 508

As I also stated in a major comment, if my interpretation is correct, your results and the results of the re-analysis of Witek et al. (2014) together indicate that there is an inverted-U relationship between the urge to move and rhythmic complexity in general. However, if we only look at idiomatic complexity, there is no such relationship because the range of complexity is narrow. It is plausible to find a null effect of complexity if we assume that the complexities of the rhythms used in this study fall within the optimal range. I believe that the results of this study (and the re-analysis of Witek et al. 2014) reflect that renowned drummers are able to choose optimal rhythmic complexity; they know that people don't prefer rhythms that are too simple or too complex.

10. line 517

I think that if you want to cite them, you should wait for them to be published. Please check the PLOS ONE guidelines once again.

11. line 526

"predicotr" → "predictor"

12. line 542

Please also check if "Kawase et al., in press" can be cited.

6. PLOS authors have the option to publish the peer review history of their article (what does this mean?). If published, this will include your full peer review and any attached files.

Reviewer #1: **Yes: **Rainer Polak

Reviewer #2: No

Reviewer #3: **Yes: **George Sioros

Reviewer #4: No

---

## [Author Response · Author response to Decision Letter 0]

31 Aug 2024

Dear Reviewers, dear PLOS One Academic Editor

Many thanks for engaging with our study so closely and for offering detailled, thorough, and constructive commentary on so many aspects of the paper. This is peer-review at its best: we highly appreciate the effort that you put in the reviews, and we don’t take it for granted. 

We respond in kind by re-submitting the paper with many major and minor revisions; and we hope the new version addresses your concerns in a satisfactory way. You’ll find our direct response to your critique and recommendations in this rebuttal letter below, where line numbers (L…) point to changes in the manuscript.

The most substantial changes can be found in the Discussion section: large parts were rewritten from scratch in order offer a more in-depth engagement with our own results and to better connect them to (or contrast them with) other findings in the field.

We appreciate that you take the time to engage with our study yet another time, and we are looking forward to your comments and further criticism.

Kind regards

Olivier Senn & Co-authors

Reviewer 1

The study is relevant and interesting. It is very well crafted and presented. 

Thank you for your appreciation of the study in general and for your very motivating comments – we appreciate it!

“Why was the inverted-U hypothesis not confirmed in this study?” Unfortunately, the biggest portion of this section does not answer this question but rather discusses why earlier studies might have found it in contrast to the present one. […] It reads almost as if they wanted to explain away their main finding rather than discuss it.

Upon re-reading the Discussion section of our first submission we agree with you that this section was too superficial. It lacked a coherent argument that connects our study with past research, and we obviously shied away from drawing inconvenient conclusions about the inverted-U hypothesis.

To address your critique, we rewrote the entire first part of the Discussion in L531-655. The new version takes a closer look at the stimulus set of our own study and of other studies that investigate the inverted-U hypothesis. We argue that the complexity of the stimuli in most of the replications is confounded with the different levels of ecological validity among the stimuli: Medium-complexity stimuli tend to be based on idiomatic musical patterns created by musicians, while, in many studies, most of the low and high complexity stimuli are based on potentially un-idiomatic patterns created by researchers. We suggest that the inverted-U relationship between complexity and the urge to move found in many studies may be an effect of using idiomatic and un-idiomatic stimuli in the same experiments. We hypothesize that – in accordance with our results – there may not be a systematic relationship between the complexity of popular music drum patterns and their capacity of motivating listeners to move, as long as we only consider idiomatic patterns with high ecological validity. With your permission, we gladly use your observations on the existence of simple and complex dance music and the role of familiarity to support this point (L643-647).

Let us know to what extent the rewrite solves the problems that you identified in the first part of the Discussion section. The section now runs a bit long and potentially presents too much detail. This is not a serious problem, because PLOS does not impose an upper word count limit, yet conciseness is a valued property nevertheless. If you see parts that can safely be shortened, we are grateful for your advice.

The authors mention that a possible limitation of the study is that the stimuli were modeled after performances by highly competent musicians. In the same vein, they stated earlier in the discussion that “Great drummers can play a simple pattern in such a way that it is still interesting to the listener, and they ca [sic] present a complex pattern such that listeners still feel the regularity.” I am skeptical about this representing a limitation.

The reviewer observes that this point was mentioned twice, once in the Discussion and another time in the Limitations sections. We omitted the first mention of this point in the Discussion and made it more concise in the Limitations (L718-728). In general, we concur with the reviewer that the performance skill of the performer is not likely to affect the results, since the reproduction only captures a few aspects of the individual nuances of the actual performance. We removed this point from the text, since we cannot offer a valid assessment of how close the reproductions are to the original performances (other than relating that we tried to make them as good as possible, L177-194). Nevertheless, we still think there might have been some kind of selection bias: choosing only renowned drummers and famous/iconic patterns may have an effect onthe relationship between complexity and the urge to move. We tried to clarify this thought in L718-L728.

The authors then mention as a limitation that the stimuli cover a range of complexity that is limited to the range that is typical of the represented genres. Again, I do not think this is a limitation of the study but rather adds to its ecological validity.

Agreed. We left this as a limitation (L718-L728), but focused more on the aspect that we used mainstream popular music and did not consider more experimental genres.

Alongside the above limitations relating to the details of the auditory stimuli used in the present study, it might also be worth going one level of abstraction higher and reflecting on the concept of psychological groove research a bit more generally. The following limitations concern the present study and the field at large.

Reductionism is a generalized problem in groove research: stimuli in experiments are usually thinned, shortened and simplified shadows of what music is in the real world. The repertoire is often conventiently restrained to specific genres, and the experimental listening situations often have little in common with the ways people usually experience music (with a few notable exceptions in which groove research managed to use real music in a real listening situation, such as Swarbrick et al, 2019; Cameron et al., 2022). Reductionism not only affects the stimuli (drum set vs. full band) and musical styles (rock, pop, funk vs. everything else), but also the perception modalities (auditory only vs. multimodal) and the listening situation (filling a questionnaire at a computer vs. moving freely to music in a real-world situation). We took the liberty to pack all your points about the problems of reductionism into one big limitation (L746-761). We are aware of these problems, and it does not hurt to state them explicitly. 

The groove model framework and its implementation with scales and SEM etc. lead to a fairly complicated data collection procedure, which, unfortunately, is not improving the listening experience for the participants (stimuli in small chunks, lots of repetitions, lots of ratings). Thus the model adds to the problem that the online survey situation is not a natural listening situation – we acknowledge this in L739-745. We just hope that the perks of the model (simultaneous testing of a constellation of direct and indirect effects) at least allows to better contextualise the groove experience in the long run and improve our understanding of the phenomenon.

Smaller points: 

Danceability: We introduced danceable as a short form of “which motivates dancing”. But we agree that this might be a bit confusing, so we replaced the expression with longer, but hopefully clearer formulations (L17, L18).

L231: “the music wit” � “musical stimulus with”. Corrected in L267.

L233: “to what extent feel” � “to what extent listeners feel”. Corrected in L273.

L477: “they ca represent” � “they can represent” Sentence removed during rewrite.

L526: “predicotr” � “predictor”. Corrected in L672.

Reviewer 2

Many thanks for your excellent suggestions. We have made the following changes to the manuscript in order to address your comments:

1) The authors did not test (or show) the relationship between syncopation and MOV level for their stimuli.

Thank you for highlighting the Syncopation � MOV relationship. We had a short passage about this aspect in the originally submitted manuscript, but we gladly present it a bit more prominently. We have added more detail in the text (L408-L429) plus a new figure (Figure 3), which shows that this relationship turns out to be “U”, but not “inverted-U”.

Additionally, I wonder if the range of syncopation in the dataset is too narrow to reveal a clear inverse U-shaped curve.

We added information about the range on the Index of Syncopation in L406-409. Note that we had to change this metric to Syncopation per beat in order to make the syncopation values of the two studies compatible with each other (Witek et al.’s patterns have a length of 2 bars, whereas the patterns in the current new study have a length of 4 bars), so simply summing up syncopation values would be misleading . 

It turns out that the range of the Syncopation per beat values in Witek et al. (2014) of [0.0, 10.1] (L414) are indeed 30% larger than the range in the new study [0.5, 8.3] (L412). If we only consider drummer-composed patterns, then the range of syncopation values of Witek et al. is [0.875, 8.875] and thus quite similar to the range in the new study.

Note that we reframed the argument in the first part of the Discussion section (L531-655) in order to better explain our take on the differences between the replication studies that confirmed the inverted-U and our new study. Our main point is that previous replications use idiomatic musical patterns in the mid-complexity range, and newly composed, potentially unidiomatic patterns in the low- and high-complexity domains. This also addresses the complexity range, since the un-idiomatic patterns are situated at the extremes of the syncopation dimension. 

2. It would greatly enhance the readability of the article to better explain this measure and how it diverges from the level of syncopation in the methods section.

We agree completely on this point: we added a minimal set of explanations on the complexity measure in L195-237, and directing the reader to Senn et al. (2023a) for more more detailled information. Let us know if this is sufficient or whether more detail should be given.

3. Perceived complexity seems to be a subjective measure of musical stimuli. Could the absence of an inverse U-shaped relationship between this metric and MOV be due to potential covariations?

We are not entirely sure whether we understand your question correctly. We agree that the Syncopation Index is an objective measure (i.e. it only depends on the properties of the drum pattern itself), and perceived complexity is a subjective measure (i.e. it depends on the perception and judgement of people hearing a pattern). The two measures are positively correlated with each other in the case of our stimuli (r=0.7), but they are not identical. Our data did not confirm the inverted-U-hypothesis in either case, regardless of whether we regressed MOV on the Index of Syncopation or on Perceived Complexity.

What do you mean by covariations? Do you think of any specific co-variate that we should consider and that, if included, might lead to a different result? If so, please elaborate which co-variates might be interesting to test. 

Minor Revisions

1. “the music wit” � “the music with”: corrected (L267).

2. “predicotr” � “predictor”: corrected (L672).

3. “SEM”: thank you for catching this, we have added the complete formulation to the first appearance of the acronym (L123).

Reviewer 3

First of all, I would like to thank the authors for their excellent paper, which I have read with great interest and pleasure.

Many thanks for your kind words and constructive comments!

This result is very similar to that reported by Sioros et al 2022, where the original syncopated patterns of commercial songs were rated higher than algorithmically syncopated patterns of the same songs.

Thank you for (re-)directing our attention to the Sioros et al. (2022) study. While preparing our experiment, we were aware of this study, but we did not yet understand how much its design and results help with the interpretation of our results. We have rewritten large portions of the Discussion session (L531-655) in order to give a clearer interpretation of the results (also in response of the remarks of other reviewers). In the course of this rewrite, we also added a paragraph (L605-636) on the Sioros et al. (2022) study: this allows us to put the entire discussion on idiomatic vs. un-idiomatic patterns into perspective.

[…] I wonder if some analysis regarding cross-rhythms and pickups could also be done in the Witek et al. (2014) dataset.

Indeed, it would be very interesting to carry out this kind of analysis. This might start with the statistical analysis of a larger corpus (e.g. the drum patterns published by Hosken et al., 2021) that would offer a great basis for the description of syntax. Concrete musical patterns from stimuli set such as Witek et al., (2014) could then be discussed with respect to the syntax. Given that this would be a major enterprise and that the current paper is already very long, we refrain from doing any type of analysis in this direction. However, we suggest it should be done in the future (L632-636).

2) I find that lines 8-10 of the abstract are not very clear as to what the novel contribution of this paper is.

We agree, this was not yet to the point. We tried to formulate this more clearly (L6-14). What do you think?

3) A reference to a definition of syncopation may be needed in lines 80-82.

We added references and adapted our definition of syncopation in order to better link the references and PCRI (L74-80).

4) But doesn't this mean that other features of music are actually more important than complexity for our urge to move?

We agree entirely: there certainly are other aspects that are important for the groove experience, besides complexity (which potentially is not very relevant at all, see L640-643). In the mentioned passage of the first submissien we intended to point to performative aspects that are likely to be relevant. Note that this passage was removed in the course of the rewrite, but some ideas can be found in the Limitations section (L718-728).

5) Syncopation is not linearly related to complexity, even though complexity and syncopation have sometimes been used almost interchangeably in the literature. […] I do not disagree with the specific lines in the abstract, but perhaps this should be further clarified in the text […]

We have added a critique of syncopation as a measure of complexity already in the abstract (L8-10). Mentioning that syncopation is not a perfect measure of complexity added to our motivation of carrying out yet another replication with new stimuli and a perceptual measure of complexity. We also reformulated the corresponding passage in the Introduction to better differentiate syncopation and complexity (L65-L70).

6) line 510-512: This is an excellent result! I would even remove the word "every-day". The experience of groove is genre specific, even if there may be some universal principles behind it.

Thank you! We slightly reformulated this point in the course of the rewrite of the Discussion. “Every-day” came from the idea of contrasting the lab situation with listening situations in the real world that should be the focus of our investigations. We tried to make this contrast a bit stronger (L746-761).

“they ca represent” � “they can represent”: Thank you for catching this, the sentence was removed in the course of rewriting the Discussion section.

Reviewer 4

Firstly, I like this study very much. The study has clear hypotheses and appropriate methods of analyses. Additionally, the results are interesting and advance our understanding of the relationship between the groove experience and rhythmic complexity. 

Many thanks for your kind assessment of the study as a whole, and especially for your constructive and helpful com

---

## [Decision Letter · Decision Letter 1]

27 Sep 2024

Null effect of perceived drum pattern complexity on the experience of groove

PONE-D-24-17648R1

Dear Dr. Senn,

We’re pleased to inform you that your manuscript has been judged scientifically suitable for publication and will be formally accepted for publication once it meets all outstanding technical requirements.

Kind regards,

Phakkharawat Sittiprapaporn, Ph.D.

Academic Editor

PLOS ONE

Reviewers' comments:

Reviewer's Responses to Questions

**Comments to the Author**

1. If the authors have adequately addressed your comments raised in a previous round of review and you feel that this manuscript is now acceptable for publication, you may indicate that here to bypass the “Comments to the Author” section, enter your conflict of interest statement in the “Confidential to Editor” section, and submit your "Accept" recommendation.

Reviewer #1: All comments have been addressed

Reviewer #2: (No Response)

Reviewer #4: All comments have been addressed

2. Is the manuscript technically sound, and do the data support the conclusions?

Reviewer #1: Yes

Reviewer #2: Yes

Reviewer #4: Yes

3. Has the statistical analysis been performed appropriately and rigorously? 

Reviewer #1: I Don't Know

Reviewer #2: Yes

Reviewer #4: Yes

4. Have the authors made all data underlying the findings in their manuscript fully available?

Reviewer #1: Yes

Reviewer #2: Yes

Reviewer #4: Yes

5. Is the manuscript presented in an intelligible fashion and written in standard English?

Reviewer #1: Yes

Reviewer #2: Yes

Reviewer #4: Yes

6. Review Comments to the Author

Reviewer #1: (No Response)

Reviewer #2: I’m happy with the changes that the authors made to the revised manuscript in response to the

reviewers’ comments including my own. I have no more suggestions.

Reviewer #4: (No Response)

7. PLOS authors have the option to publish the peer review history of their article (what does this mean?). If published, this will include your full peer review and any attached files.

Reviewer #1: **Yes: **Rainer Polak

Reviewer #2: No

Reviewer #4: No

---

## [Editor Report · Acceptance letter]

11 Oct 2024

PONE-D-24-17648R1 

PLOS ONE

Dear Dr. Senn, 

I'm pleased to inform you that your manuscript has been deemed suitable for publication in PLOS ONE. Congratulations! Your manuscript is now being handed over to our production team.

Kind regards, 

on behalf of

Dr. Phakkharawat Sittiprapaporn 

Academic Editor

PLOS ONE